# The context-dependent epigenetic and organogenesis programs determine 3D vs. 2D cellular fitness of MYC-driven murine liver cancer cells

Jie Fang[1†], Shivendra Singh[1†], Brennan Wells[1], Qiong Wu[1], Hongjian Jin[2], Laura J Janke[3], Shibiao Wan[4], Jacob A Steele[5], Jon P Connelly[5], Andrew J Murphy[1], Ruoning Wang[6], Andrew Davidoff[1,7,8], Margaret Ashcroft[9], Shondra M Pruett-Miller[5], Jun Yang[1,7,8,10]*

[1]Department of Surgery, St Jude Children's Research Hospital, Memphis, United States; [2]Center for Applied Bioinformatics, St Jude Children's Research Hospital, Memphis, United States; [3]Department of Pathology and Division of Comparative Pathology, St. Jude Children's Research Hospital, Memphis, United States; [4]Bioinformatics and Systems Biology Core and Department of Genetics, Cell Biology and Anatomy University of Nebraska Medical Center, Omaha, United States; [5]Department of Cell and Molecular Biology, Center for Advanced Genome Engineering, St. Jude Children's Research Hospital, Memphis, United States; [6]Center for Childhood Cancer Research, Hematology/Oncology & BMT, Abigail Wexner Research Institute at Nationwide Children's Hospital, Department of Pediatrics at The Ohio State University, Columbus, United States; [7]St Jude Graduate School of Biomedical Sciences, St Jude Children's Research Hospital, Memphis, United States; [8]Department of Pathology and Laboratory Medicine, College of Medicine, The University of Tennessee Health Science Center, Memphis, United States; [9]Department of Medicine, University of Cambridge, Cambridge, United Kingdom; [10]College of Graduate Health Sciences, University of Tennessee Health Science Center, Memphis, United States

*For correspondence:
Jun.Yang2@stjude.org

[†]These authors contributed equally to this work

## eLife Assessment

This manuscript provides potentially **important** findings examining in 2D and 3D models in MYC liver cancer cells changes in DNA repair genes and programs in response to hypoxia. The authors use **convincing** methodology in most cases, but there is some concern that the analysis is **incomplete**.

**Abstract** 3D cellular-specific epigenetic and transcriptomic reprogramming is critical to organogenesis and tumorigenesis. Here, we dissect the distinct cell fitness in 2D (normoxia vs. chronic hypoxia) vs 3D (normoxia) culture conditions for an MYC-driven murine liver cancer model. We identify over 600 shared essential genes and additional context-specific fitness genes and pathways. Knockout of the VHL-HIF1 pathway results in incompatible fitness defects under normoxia vs. 1% oxygen or 3D culture conditions. Moreover, deletion of each of the mitochondrial respiratory electron transport chain complex has distinct fitness outcomes. Notably, multicellular organogenesis signaling pathways including TGFβ-SMAD, which is upregulated in 3D culture, specifically constrict

the uncontrolled cell proliferation in 3D while inactivation of epigenetic modifiers (*Bcor*, *Kmt2d*, *Mettl3*, and *Mettl14*) has opposite outcomes in 2D vs. 3D. We further identify a 3D-dependent synthetic lethality with partial loss of *Prmt5* due to a reduction of *Mtap* expression resulting from 3D-specific epigenetic reprogramming. Our study highlights unique epigenetic, metabolic, and organogenesis signaling dependencies under different cellular settings.

## Introduction

Cellular fitness is determined by a constellation of genetic, epigenetic, metabolic, and environmental factors that dynamically interact and communicate (*Kaelin, 2005*). Identifying the genes and pathways controlling cellular fitness not only enhances understanding of the pathophysiological mechanisms, but also helps to define potential therapeutic targets to treat human disorders. Genome-wide CRISPR and shRNA screenings are powerful genetic approaches that are invaluable for identifying fitness genes (*Tsherniak et al., 2017*; *McDonald et al., 2017*; *Dharia et al., 2021*; *Sun et al., 2023*; *Behan et al., 2019*). While some studies have applied genome-wide or focused CRISPR library screenings using 3D organoids (*Ungricht et al., 2022*; *Murakami et al., 2021*; *Ringel et al., 2020*; *Takeda et al., 2019*; *Han et al., 2020*), which better recapitulate in vivo tissue features, most genome-wide screenings have been performed using 2D cultured cells grown in normoxia (normoxia or normoxic conditions tested refer to the standard ambient air oxygen concentrations used in conventional in vitro models). However, direct comparison of cell fitness in 3D vs 2D culture is currently lacking. Two recent studies have identified several hundred oxygen concentration-dependent fitness genes (*Thomas et al., 2021*; *Jain et al., 2020*), demonstrating that environmental factors such as oxygen have a profound impact on cellular fitness. We previously showed that organoids in normoxic culture induced a strong hypoxia signature in cells (*Yang et al., 2009*), indicating that 3D culture may mimic 2D hypoxic culture, at least to some degree. Oxygen is an essential molecule that fostered the evolution of monocellular to multicellular organisms. Oxygen is primarily utilized by mitochondria as the final electron acceptor to generate ATPs in the oxidative phosphorylation pathway, as well as a variety of chemical reactions in cells (*Gan and Ooi, 2020*; *Thannickal, 2009*; *Lee et al., 2020*). Physiological oxygen concentrations can vary anywhere from 0.5% in the large intestine to just under 21% in the trachea, with most major organs operating around 3–7% oxygen (*Jain et al., 2020*; *Yang et al., 2009*; *Gan and Ooi, 2020*). To maintain cellular fitness, cells have evolved oxygen-sensing mechanisms to adapt oxygen variation by altering expression of a plethora of genes, mainly through hypoxia-inducible factors (HIFs) and epigenetic modifiers such as histone demethylases some of which are HIF targets (*Lee et al., 2020*; *Kaelin and Ratcliffe, 2008*; *Chakraborty et al., 2019*). HIF is a heterodimer composed of HIF-1α or HIF-2α (also named EPAS1) and their partner HIF-1β (also known as ARNT; *Kaelin and Ratcliffe, 2008*). While HIF-1β is constitutively expressed, HIF-α availability is tightly regulated by prolyl hydroxylases (PHD) and the von Hippel Lindau (VHL) tumor suppressor protein (*Kaelin and Ratcliffe, 2008*). At ambient oxygen levels, HIF-α is hydroxylated by oxygen-dependent PHDs leading to its rapid proteasomal degradation by the VHL E3 ubiquitin ligase complex. Conversely, hypoxic conditions render HIF-α available for dimerization with HIF-1β. The HIF-α/HIF-1β complex then translocate to the nucleus, binds to the hypoxia response elements (HRE), and drives gene expression. The dysregulation of HIF-PHD-VHL pathway is associated with many human disorders including cardiovascular diseases and cancers (*Liang et al., 2023*; *Singleton et al., 2021*).

Organogenesis is an orchestrated process that is regulated by developmental signaling pathways mediated by TGFβ, Wnt, Hedgehog, Notch, and others. Genetic alterations in these signaling pathways cause cancers and other human diseases (*Pelullo et al., 2019*). Solid tumors are recognized as functionally abnormal organs (*Egeblad et al., 2010*), and they are classically under a heterogeneous state of hypoxia in part due to insufficient and pathologic angiogenesis (*Harris, 2002*). The dynamic cell-cell and cell-environment interactions of tumor cells in 3D suggest an added layer of complexity to our current understanding of tumor biology (*Egeblad et al., 2010*). Clonal monoculture studies do not fully capture the hypoxia- and three-dimensional-driven mechanisms found in heterogeneous 3D tumors (*Egeblad et al., 2010*). Single-cell RNA-seq analysis of 1163 human tumor samples covering 24 tumor types shows that hypoxia, together with *MYC* gene clusters are the two of the most commonly recurring transcriptional programs in heterogenous tumors (*Gavish et al., 2023*). Identifying fitness

genes of MYC-driven cancers in different oxygen levels and in 3D conditions may reveal context-dependent vulnerabilities that can be exploited for therapies.

Here, we show that under 3D culture conditions, cells undergo unique epigenetic and transcriptomic reprogramming of gene transcription, particularly for the genes regulated by TGFβ-SMAD signaling. Further, we use genome-wide CRISPR/Cas9 gene-editing technology to knock out every gene in an MYC-driven cancer cell line cultured as a monolayer under normoxia and 1% oxygen, or in 3D spheroids under normoxia, followed by comprehensive comparative analyses of cellular fitness genes under these three conditions. We show that knockout of *Vhl* leads to a fitness defect in normoxic 2D conditions but not in 1% oxygen monolayer and 3D spheroids. Conversely, deletion of *Hif1a* or *Hif1b* causes fitness defects in 1% oxygen and 3D spheroids but not in a normoxic 2D conditions. Cells tolerate loss of mitochondrial ribosomal genes better in 1% oxygen than normoxia. However, knockout of mitochondrial respiratory complex I-V causes distinct cellular fitness outcomes for each complex under these three conditions. Knockout of genes in organogenesis signaling pathways such as TGFβ-SMAD specifically leads to a growth advantage of 3D spheroids while inactivation of epigenetic modifiers (*Bcor*, *Kmt2d*, *Mettl3,* and *Mettl14*) in monolayer and 3D spheroids results in opposite fitness outcomes, and these genes often function as tumor suppressors in human cancer. We also discover distinct metabolic requirements for fatty acid and cholesterol synthesis in the three conditions. We further identify a context-dependent synthetic lethality of 3D culture conditions with partial loss of *Prmt5* due to a reduction of *Mtap* expression, resulting from 3D-specific epigenetic reprogramming of the *Mtap* gene. Our study reveals distinct epigenetic and metabolic dependencies of cancer cells in different environments, highlighting the critical role of organogenesis signaling pathways in regulating tumor cell growth.

## Results

### Transcriptomic reprogramming of cancer cells under 1% oxygen in 2D or normoxic 3D culture conditions

We have recently generated an MYC-driven liver cancer genetic model (ABC-Myc mouse) and derived cell lines such as NEJF10 (*Fang et al., 2023*), which are readily cultured in 2D and 3D conditions (*Figure 1—figure supplement 1A*), and which are suitable for genome-wide genetic screenings (*Fang et al., 2023*). In comparison with human cancer cell lines which usually have multiple genetic defects, the ABC-Myc cell lines have a simpler genotype (MYC alone as a driver), which may be less impacted by epistasis when used for genome-wide identification of fitness genes. To understand the gene transcription differences in normoxia vs hypoxia, and 2D vs 3D culture, we performed RNA-seq after NEJF10 cells were cultured in normoxia and 1% oxygen in monolayer, or normoxic 3D spheroids for 48 hr. We performed pairwise comparisons of genes upregulated and downregulated in monolayers grown under hypoxic (1%) or normoxic (21%) conditions to those of a culture grown as 3D spheroids under normoxia (21%). We found that 44.8% (925 out of 2065) and 49.4% (1021 out of 2065) of genes that were upregulated and downregulated under 1% oxygen were shared with the culture grown as 3D spheroids, respectively (*Figure 1—figure supplement 1B*, *Supplementary files 1 and 2*). The most significant gene signatures upregulated in 3D vs 2D under 21% oxygen are hypoxia-related (*Figure 1—figure supplement 1B*), consistent with our previous observation that organoid culture induced a strong hypoxia signature (*Yang et al., 2009*). REACTOME and KEGG pathway enrichment analysis revealed that hypoxia and 3D culture induced common genes involved in focal adhesion, proteoglycans, extracellular matrix and receptor interaction, as well as oncogenic signaling pathways such as Hippo, HIF1, Wnt, and PI3K-AKT (*Sanchez-Vega et al., 2018*), but inhibited the expression of genes involved in mitochondrial translation and the proteasome (*Figure 1C*, left; *Figure 1—figure supplement 1C*). However, genes involved in protein translation or ribosome biology, oxidative phosphorylation and protein export were dominantly downregulated by hypoxia (*Figure 1C*, right; *Figure 1—figure supplement 1C*), while 3D culture enhanced the expression of genes in TGF-β-SMAD signaling, regulation of pyruvate dehydrogenase (PDH) complex, NOTCH, MAPK, autophagy, and lysosome pathways (*Figure 1C*, middle; *Figure 1—figure supplement 1C*). 3D culture also suppressed the expression of genes in the DNA homologous recombination and Fanconi anemia pathway (*Figure 1C*, middle; *Figure 1—figure supplement 1C*), which are involved in DNA repair and frequently mutated in various cancers (*Knijnenburg et al., 2018*).

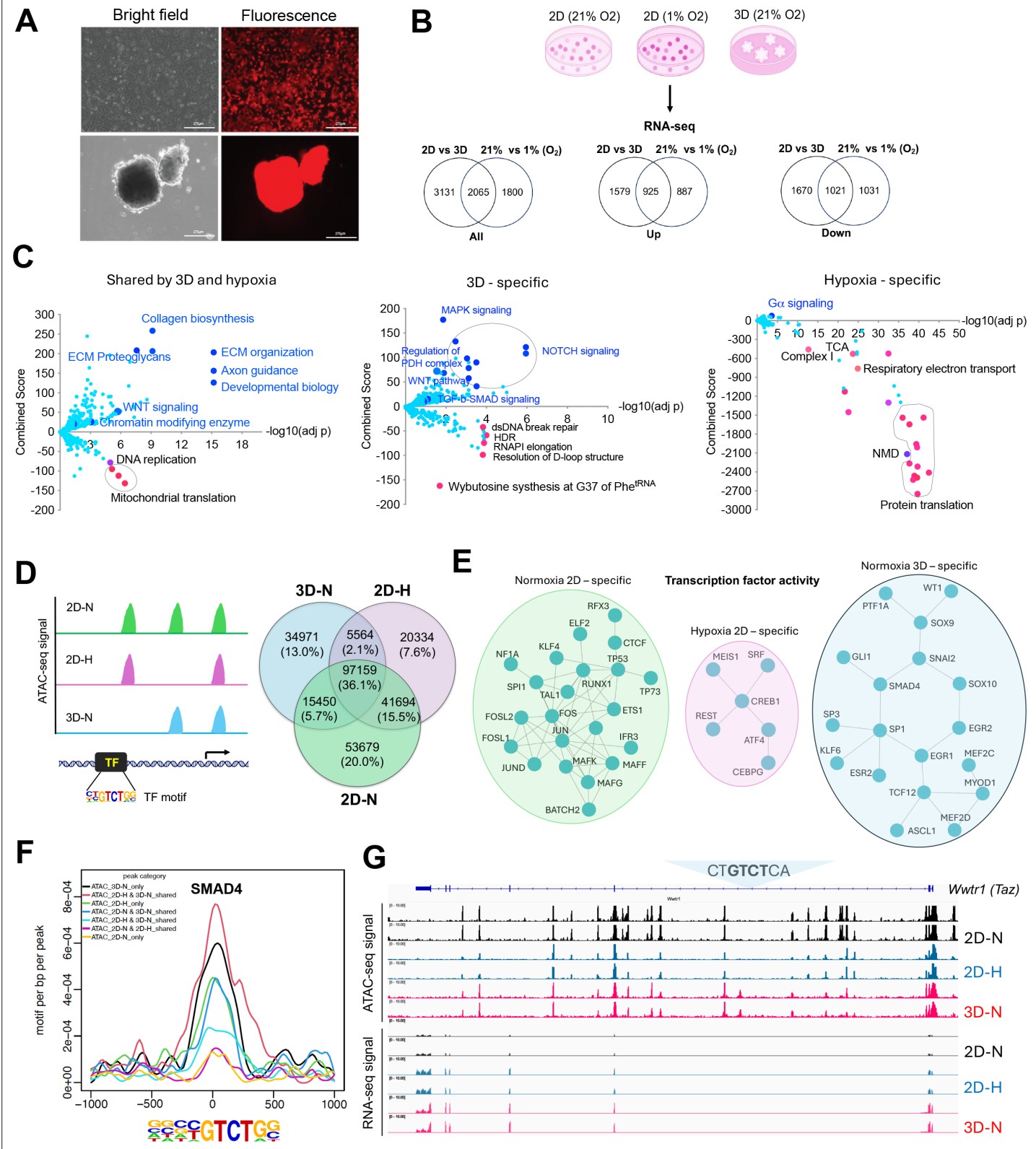

**Figure 1.** Transcriptomic and epigenetic reprogramming in hypoxic 2D and normoxic 3D conditions. (**A**) The ABC-Myc-driven hepatoblastoma organoids serve as a good model for genome wide fitness screening. The ABC-Myc lineage was validated through expression of TdTomato. scale bar = 275μm (**B**) RNA-sequencing data from cells cultured in our conditions of interest are represented in Venn Diagrams of genes in 3D vs 2D normoxia and 2D-Hypoxia vs 2D-Normoxia (n=3 per group). The leftmost Venn diagram represents a total count of all genes identified, followed by the Up and

*Figure 1 continued on next page*

*Figure 1 continued*

down labeled diagrams reporting the upregulated and downregulated genes, respectively. The numbers in the Venn diagrams represent the number of significant genes expressed in the respective conditions. Created with BioRender. (**C**) The volcano plots, generated using Enrichr, show significant REACTOME pathways shared by cells in normoxic 3D and hypoxic 2D (left), specific to hypoxic 2D (middle) and normoxic 3D conditions (right; n=3 per group). The X- axis represents the significance of expression change for a gene in –log10 (adjusted p- value). The Y- axis represents the combined score for enrichment of a given pathway with positive values indicating enrichment and negative values indicating a decrease in the pathway. (**D**) ATAC-seq analysis for cells under normoxic 2D, hypoxic 2D, and normoxic 3D conditions (n=2 per group). The left cartoon indicates the changes of ATAC-seq signals under different culture conditions and predicted transcription factor binding motif. Right Venn diagram showing the common and unique ATAC-seq signals (numbers and percentages) under three conditions. (**E**) Protein-protein interaction network (STRING confidence threshold = 0.7) formed by transcription factors that are predicted to be highly active under normoxic 2D, hypoxic 2D, and normoxic 3D, respectively. (**F**) SMAD4 motif densities around ATAC-seq open chromatin regions (±1000 bp) by categories. Motif density was determined by HOMER (Hypergeometric Optimization of Motif EnRichment) program and normalized to that in a background sequence of equal length. X-axis indicates the distance to peak center (bp). (**G**) IGV snapshot shows the ATAC-seq signals and RNA-seq reads under normoxic 2D, hypoxic 2D, and normoxic 3D conditions. The 'CTGTCTCA' SMAD4 binding motif lies within the ATAC-seq peak specifically appears in normoxic 3D conditions.

The online version of this article includes the following figure supplement(s) for figure 1:

**Figure supplement 1.** Cells undergo different transcriptional programing in hypoxia, normoxia, and 3D culture.

**Figure supplement 2.** Epigenetic reprogramming under hypoxic 2D and normoxic 3D conditions.

**Figure supplement 3.** HIF motif densities surrounding the ATAC-seq peaks.

Alternative splicing is an important mechanism cells use to adapt to cellular stresses by producing different gene isoforms (*Biamonti and Caceres, 2009*). We therefore compared the five major splicing events (alternative 3' splicing, alternative 5' splicing, mutually exclusive exon usage, intron retention and exon skipping) induced by hypoxia and 3D culture (*Supplementary files 3-5*). In comparison with normoxia in 2D culture, hypoxia dominantly induced intron retention, while 3D cultures dominantly induced exon skipping (*Figure 1—figure supplement 1D*). Exon skipping was also the major event in 3D culture when compared with 2D hypoxic cultures (*Figure 1—figure supplement 1D*). These data indicate that splicing is an important mechanism for cellular adaptation to environmental stress, as evidenced by the altered splicing events of pyruvate dehydrogenase kinase 1 (*Pdk1*) gene under different culture conditions (*Figure 1—figure supplement 1E*). Exon skipping in exons 7 and 8 of *Pdk1* specifically occurred in 3D culture, while an alternative 5' splicing event only occurred in hypoxic conditions (*Figure 1—figure supplement 1E*). Pathway analysis of the five splicing events under three conditions demonstrated that various biological processes (i.e. ribosome, spliceosome, metabolic pathways, and DNA repair) are involved in all conditions (*Figure 1—figure supplement 1F*, *Supplementary files 6-8*), indicating an overly complex regulation of post-transcriptional processing. Taken together, these data demonstrate that cells undergo commonly shared and context-dependent transcriptional reprogramming in normoxia, hypoxia, and 3D spheroid conditions. Particularly, pathways involved in multicellular organogenesis, tissue homeostasis, and a variety of human diseases are specifically induced by hypoxia and/or 3D culture, suggesting that conventional 2D culture under normoxia may not recapitulate the cellular fitness in 3D culture or under hypoxic conditions.

## Context-specific epigenetic reprogramming of cells in hypoxic 2D and normoxic 3D conditions

To determine the mechanism by which culture conditions induced context-specific transcriptomes, we performed an assay for transposase-accessible chromatin with sequencing (ATAC-Seq) for determining chromatin accessibility across the genome in normoxic and hypoxic 2D and normoxic 3D culture conditions. The results showed a drastic effect of culture conditions on DNA accessibility (*Figure 1D*). While 36.1% of chromatin accessibility regions were shared in three conditions, unique chromatin accessibility was observed (20% for 2D normoxia, 7.6% for 2D hypoxia and 13% for 3D). Nevertheless, the chromatin accessibility peaks showed similar distributions at transcription start sites, intragenic regions, upstream and downstream of gene regions (*Figure 1—figure supplement 2A*). Then we performed Homer motif analysis to predict the activity of transcription factors in these conditions. Pairwise comparisons (normoxic 2D vs. hypoxic 2D, normoxic 2D vs. normoxic 3D, hypoxic 2D vs. normoxic 3D) of transcription factors followed by protein-protein interaction network analysis demonstrated that unique transcription factor modules functioned under these conditions (*Figure 1E*, *Figure 1—figure supplement 2B–D*). While AP1 (FOS, FOSL1, FOSL2, JUN, JUND) and CTCF were

more active in normoxic 2D culture, ATF4 was more selectively active in hypoxic 2D conditions. ATF4 is known to be involved in regulating the integrated stress response and cell survival under hypoxic conditions (*Liu et al., 2008*). However, under normoxic 3D culture, the activity of transcription factors such as SMAD4, SOX9, GLI1, and WT1 was specifically high (*Figure 1E*, *Figure 1—figure supplement 2B–D*). The high transcriptional activity of SMAD4 under 3D (*Figure 1F*) was consistent with the upregulation of target genes of TGF-β-SMAD signaling in normoxic 3D culture (*Figure 1C*, middle). *Wwtr1*, encoding a transcriptional cofactor TAZ downstream of the Hippo pathway, is a well-characterized target of TGF-β-SMAD (*Ríos-López et al., 2023*). RNA-seq results showed that *Wwtr1* was upregulated in 3D culture, in line with a unique ATAC-seq peak where lies a SMAD4 binding motif (GTCT; *Figure 1G*), suggesting that the chromatin was open to SMAD4 binding under normoxic 3D conditions. Interestingly, we did not observe significant changes of ATAC-seq peaks with HIF-1α and HIF-2α (EPAS1) binding motifs albeit the HIF-1β (ARNT) binding motif was greatly enriched under hypoxic 2D and normoxic 3D conditions (*Figure 1—figure supplement 3*). Considering that ARNT could heterodimerize with other bHLH transcription factors, these data may suggest that the chromatin accessibility for HIF-1α and HIF-2α is poised to be open in response to hypoxic stress.

## Cellular fitness genes under 21% and 1% oxygen in 2D and 3D culture conditions

Next, we sought to understand the cellular fitness in different culture conditions. Previous CRISPR screens under hypoxia were carried out for two weeks in K562 leukemia cells in suspension and 5 days for U2OS osteosarcoma cells in a monolayer (*Thomas et al., 2021*; *Jain et al., 2020*). While some cells may not tolerate chronic hypoxia in 2D culture, the ABC-Myc cell lines such as NEJF10 showed a similar doubling time over 4 weeks under normoxia and 1% oxygen tension in monolayer culture (*Figure 2—figure supplement 1A*), probably due to its greater dependence on glycolysis for ATP production (*Figure 2—figure supplement 1B*). Similarly, NEJF10 cells grew in 3D culture continuously (*Figure 2—figure supplement 1C*). To understand the cell fitness difference in normoxia, chronic hypoxia or 3D conditions, we generated a genome-wide mouse CRISPR knockout pooled library (Brie, lentiCRISPRv2), which includes 1000 control gRNAs and 78,637 gRNAs targeting 19,674 genes. Following a 36 hr puromycin selection after NEJF10 was transduced with the pooled library, cells were split into three different culture conditions: normoxia and 1% oxygen tension cultured as a monolayer and normoxic 3D spheroids (*Figure 2A*, *Figure 2—figure supplement 2*). Samples cultured as a monolayer were collected at different times points (days 1, 6, 9, 11, 17, 19, and 23 post-transduction for normoxia, days 4, 6, 11, 14, 17, 19, and 23 post-transduction for 1% oxygen), which allowed tracking of the gradual depletion or accumulation of gRNAs for a given gene KO in the two oxygen tensions. However, we only had a one-time point sample (day 28 post-infection) for 3D spheroids. The MAGeCK algorithm was used to identify differential fitness genes for each time point (*Li et al., 2014*). For monolayer culture, we defined the fitness genes as 'negative selection' or 'positive selection' if their depletion or accumulation appeared as significant in at least four different time points (*Supplementary file 9*). We identified 648 common genes in negative selection and 6 common genes in positive selection under all conditions. While the positive selection genes were basically tumor suppressor genes such as *Cdkn2a*, *Pten*, and *Ambra1*, the negative selection genes engaged in essential biological processes including RNA polymerase, DNA replication, ribosome, spliceosome, and pathways such as MYC and E2F (*Figure 2—figure supplement 3A, B*). Protein-protein interaction network analysis further demonstrated that the essential genes under all three conditions encode proteins involved in splicing, transcription, translation, proteasome, and other critical cellular functions (*Figure 2—figure supplement 3C*).

Since 3D culture itself leads to a relatively hypoxic core of cells within each spheroid and therefore partially recapitulates the transcriptional program of hypoxia pathway induction, we examined the fitness genes shared by monolayer under 1% oxygen tension and 3D under normoxia. Pathway analysis revealed that genes that engage in DNA repair and oxidative phosphorylation were essential to NEJF10 survival under both conditions (*Figure 2—figure supplement 4A, B*). While hypoxia may promote a switch from oxidative phosphorylation to glycolysis, studies showed that the low oxygen concentrations in tumors may not be limiting for oxidative phosphorylation, and ATP is generated by oxidative phosphorylation in tumors even at very low oxygen tensions (*Ashton et al., 2018*). Therefore, NEJF10 cells might be still dependent on oxidative phosphorylation for ATP production under

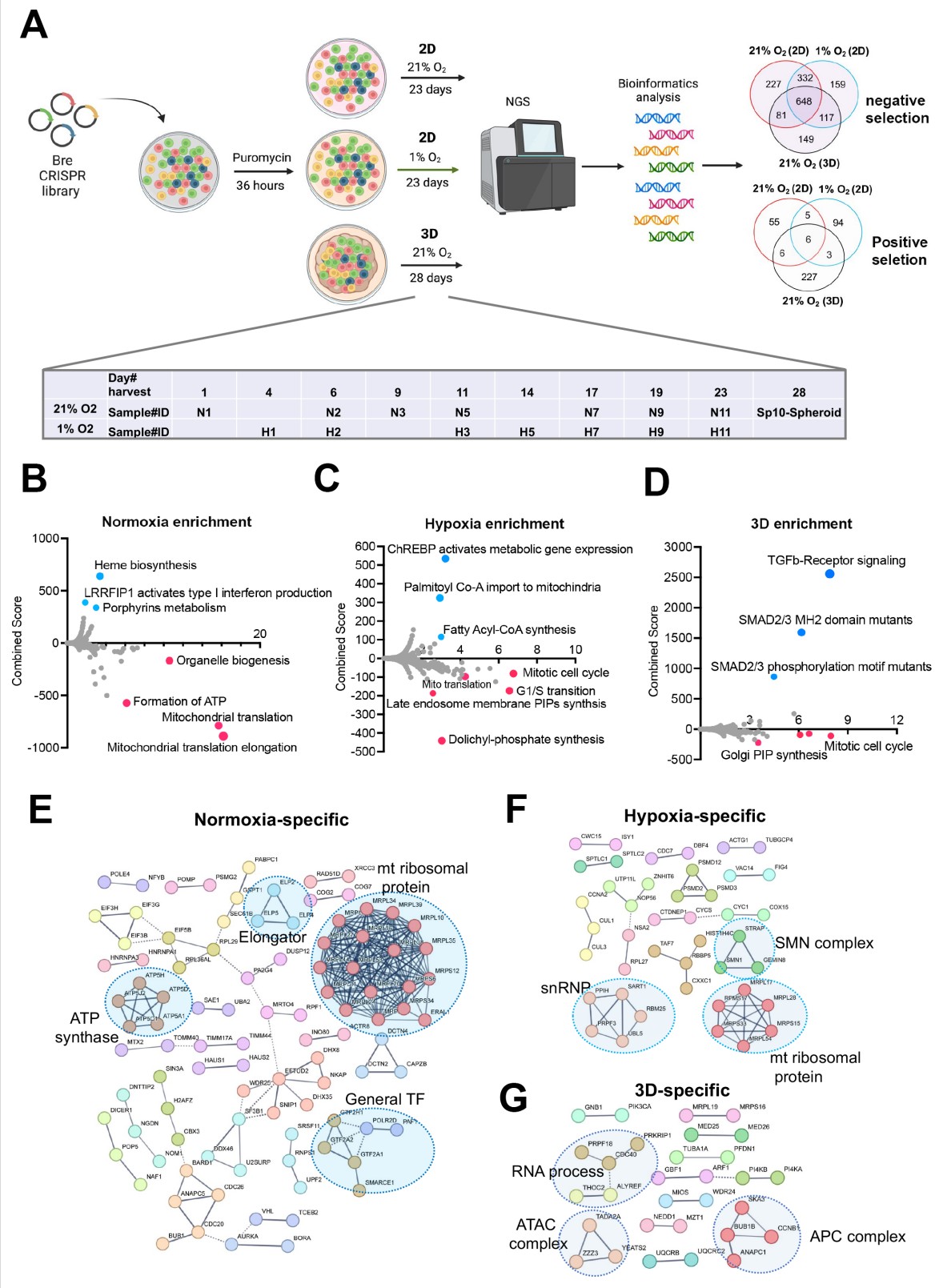

**Figure 2.** Identification of cell fitness genes in 2D hypoxia, 2D normoxia, and 3D normoxia. (**A**) Scheme showing the procedure of genome- wide CRISPR screen of the ABC-Myc NEJF10 cell line treated with different growth environments. Venn analysis of essential genes (negative selection) and anti-proliferative genes (positive selection) reveals gene essentiality for cellular fitness identified in 2D normoxia, 2D hypoxia, and 3D Normoxia. The table is a report of when samples were harvested with N representing 2D normoxia, H representing 2D hypoxia, and Sp10- Spheroid representing 3D

*Figure 2 continued on next page*

*Figure 2 continued*

normoxia grown cells. Created with BioRender.(**B, C, D**) Pathway enrichment analysis reported as an enrichment score with negative values indicating negative selection, positive values indicating positive selection, and values of zero indicating no relationship of the respective condition. The colored points highlight significant upregulation (blue) or down regulation (red), whereas grey colored points were not significantly enriched. (**E, F, G**) Pathways within a protein-protein interaction network of genes essential in cellular fitness are selectively enriched in normoxic, hypoxic, and 3D specific environments. Related genes are clustered by color with some circled to indicate functional complexes.

The online version of this article includes the following figure supplement(s) for figure 2:

**Figure supplement 1.** NEJF10 cells tolerate chronic hypoxia and more rely on ATP from glycolysis.

**Figure supplement 2.** NEJF10 spheroid growth and CRISPR procedure.

**Figure supplement 3.** Shared essential genes under normoxia 3D, normoxia 2D, and hypoxia 2D.

**Figure supplement 4.** Shared essential genes under normoxia 3D and hypoxia 2D.

hypoxia. Next, we examined the context-specific fitness genes. In monolayer culture under 21% oxygen tension, genes involved in mitochondrial translation, ATP production, and organelle biogenesis were enriched in the negative selection fraction, while genes involved in heme biosynthesis or porphyrin metabolism were enriched in positive selection (*Figure 2B*). Heme is an essential molecule for living aerobic organisms. Heme biosynthesis involves an eight-step enzymatic pathway starting in mitochondria with the condensation of succinyl Co-A from the citric acid cycle and an amino acid glycine. Inactivating mutations in heme synthesis genes define a group of diseases known as porphyria (*Bissell et al., 2017*). Recent studies revealed that acute hepatic porphyria is associated with increased risk of hepatocellular carcinoma (HCC; *Fontanellas and Avila, 2022*; *Molina et al., 2022*). In addition, under 1% oxygen, cells in monolayer culture were more specifically sensitive to the loss of genes involved in cell cycle progression, dolichyl phosphate synthesis and to a subgroup of genes in mitochondrial ribosomal translation, while genes involved in lipid metabolism (ChREBP pathway, fatty acyl-coA synthesis) were enriched in positive selection (*Figure 2C*). Interestingly, while some mitotic genes and Golgi phosphatidylinositol phosphate (PIP) synthesis genes were more essential to cells under 3D culture, the TGFβ signaling pathway was uniquely and significantly enriched in positive selection in spheroids (*Figure 2D*). Pan-cancer analysis reveals that genetic alterations in TGFβ pathway members occurred in 39% of TCGA cases, which were correlated with the expression of metastasis genes and poor prognosis (*Korkut et al., 2018*). Protein-protein interaction network analysis confirmed that mitochondrial translation genes, ATP synthase genes, and general transcription factors were more specifically important for cell survival under 21% oxygen in 2D culture (*Figure 2E*), while in 1% oxygen monolayer, RNA processing genes such as survival of motor neurons (SMN) complex and small nuclear ribonucleoprotein (snRNP) were more essential to cell survival (*Figure 2F*). The SMN complex plays an essential role in the assembly of the spliceosomal snRNPs. One study has shown that neurons with low levels of SMN are more sensitive to hypoxia and will be the ones which undergo cell death first in any population (*Hernandez-Gerez et al., 2020*), supporting the hypoxia-specific role of SMN in our screening. However, in 3D culture under 21% oxygen, a subgroup of genes involved in RNA processing, the Anaphase-Promoting Complex (APC) for mitosis, and Ada Two A containing (ATAC) complex for acetylation of histone and Cyclin A/Cdk2 (*Orpinell et al., 2010*), were important for spheroid growth (*Figure 2G*). As most prior knowledge about mammalian cell cycle progression comes from 2D cell culture experiments, it remains to be understood how cells in 3D culture proceed from G1 to S, then from S to G2/M phase.

## Cell fitness differences with VHL-HIF-1α pathway depletion under normoxia vs. 1% oxygen in 2D or 3D

Considering that the VHL-HIF-α pathway plays a central role in response to hypoxic stress (*Figure 3A*), it would be expected that knockout either of VHL or HIF-α could change the cell fitness under normoxia and hypoxia. However, the two previous screenings of K562 and U2OS cells did not show a significant effect on cell fitness in either normoxia or hypoxia after deletion of HIF-α (*Thomas et al., 2021*; *Jain et al., 2020*). Rather, deletion of *VHL* in K562 cells led to a growth advantage under normoxia (*Han et al., 2020*). In our model system, knockout of *Vhl* in NEJF10 cells led to gradual gRNA depletion in normoxia but no depletion of its gRNA under 1% oxygen (*Figure 3B*), indicating that VHL loss is incompatible with cellular fitness under normoxic conditions. Conversely, knockout of *Hif1a* or *Arnt*

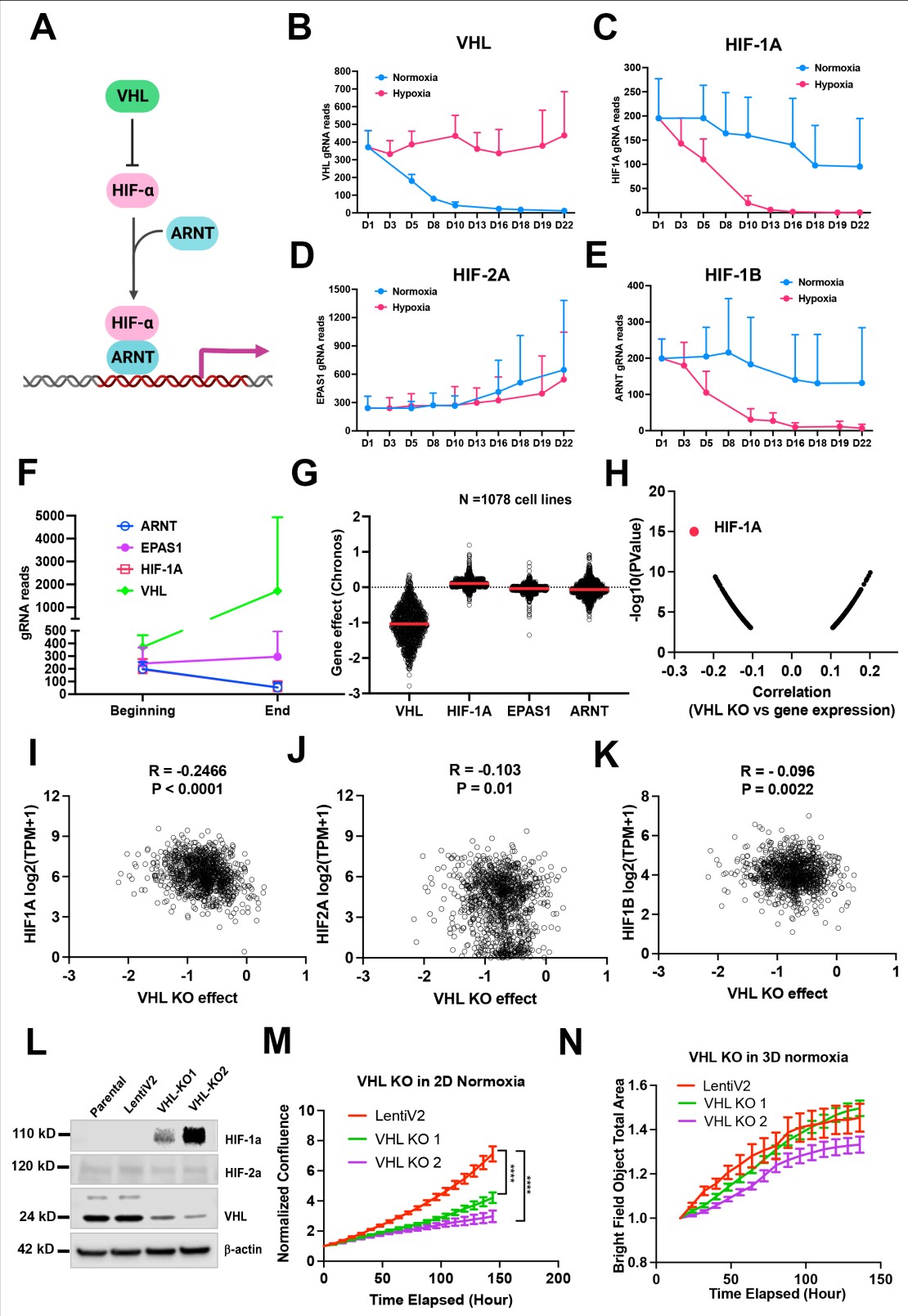

**Figure 3.** Fitness incompatibility of VHL-HIF1a pathway in normoxia vs hypoxia or 3D. (**A**) Scheme representing the VHL-HIF pathway. The HIF-1 and HIF-2a family of transcription factors are degraded by VHL, an E3 ubiquitin ligase, in normoxia. In the presence of hypoxia or loss function of VHL, HIFa is present and dimerizes with nuclear HIF1b or ARNT to drive gene transcription. (**B–E**) The gRNA reads for *Vhl* (**B**), *Hif-1a* (**C**), *Hif-2a* or *Epas1* (**D**), *Hif-1b* or *Arnt* (**E**) at different time points in 2D normoxia and 2D hypoxia. Error bar data = mean +/-SD (n=4 for each time point). (**F**) The gRNA reads for *Vhl*,

*Figure 3 continued on next page*

*Figure 3 continued*

*Hif-1a*, *Hif-2a* or *Epas1*, *Hif-1b* or *Arnt* at the beginning and end time points in 3D normoxia. (**G**) CRISPR KO effect of *VHL*, *HIF-1A*, *HIF-2A* (*EPAS1*), *HIF-1B* (*ARNT*) in 1078 human cell lines. Data were extracted from DepMAP database (http://www.depmap.org/). (**H–K**) Spearman correlation of *VHL* CRISPR KO effect vs *HIF-1A*, *HIF2A*, *HIF1B* gene expression in 1078 human cell lines. (**L**) Western blot analysis of *VHL* CRISPR knockout in HepG2 cells using two gRNAs with indicated antibodies. (**M**) Cell growth of wild-type and VHL KO HepG2 over time monitored by Incucyte live cell microscopy in real-time in 2D normoxic condition. ****p<0.0001. p value is calculated by student t test for the last time point data. Error bar data = mean +/-SD. (**N**) Cell growth of wild-type and VHL KO HepG2 over time monitored by Incucyte in real-time in 3D normoxic condition. Error bar data = mean +/-SD.

The online version of this article includes the following source data and figure supplement(s) for figure 3:

**Source data 1.** The source data contains original raw data of western blots for *Figure 3L*, labelled.

**Source data 2.** The source data contains original raw data of western blots for *Figure 3L*.

**Figure supplement 1.** Spearman correlation analysis of VHL knockout effect vs HIF.

(encoding HIF-1β) but not *Epas1* (encoding HIF-2α) led to gRNA depletion under 1% oxygen tension but not normoxia (*Figure 3C–E*). We observed similar effects of VHL-HIF pathway inactivation in 3D conditions in which cells cannot tolerate the genetic deletion of *Hif1a* and *Arnt* (*Figure 3F*). Cells in 3D even had a remarkable increase in gRNA counts of the *Vhl* gene (*Figure 3F*). By examining the DepMAP data, we found that nearly all human cancer cells cultured under 2D normoxia cannot tolerate the loss of *VHL*, while cells tended to gain growth benefit by *HIF1A* knockout under normoxic oxygen conditions (*Figure 3G*). Analysis of DepMAP data revealed that the *VHL* knockout effect was significantly negatively correlated with *HIF1A* expression levels under normoxia but had a less significant correlation with *HIF2A* and *HIF1B* (*Figure 3H–K*, *Figure 3—figure supplement 1*). We further validated the loss of function of VHL in HepG2, a human liver cancer cell line. We were unable to obtain a complete knockout of VHL in HepG2 cells when we generated stable clones that were cultured regularly in 2D under 21% oxygen (*Figure 3L*), suggesting VHL is essential to cell survival under this culture condition. Partial loss of VHL led to a significant reduction in cell proliferation in 2D but not in 3D culture under 21% oxygen tension (*Figure 3M and N*), which is in line with our screening result in NEJF10 cells. Thus, these genetic data indicate that VHL-HIF1 plays a dominant role in determining the cell fitness in normoxia and hypoxia, although cell-type-specific effects for deletion of VHL-HIF1 may also exist. Nevertheless, why some cells like U2OS were less sensitive to loss of HIF1 remains to be studied. The opposite phenotype of VHL deletion in K562 and NEJF10 cells under 21% oxygen indicates that cell-type-specific functions of VHL may be present.

## Distinct fitness outcomes for respiratory chain complex loss of function

Since mitochondrial translation and ATP synthase genes seemed to be essential for cell survival under normoxia cultured in 2D, we compared the time course of gRNA enrichment for genes with mitochondrial function. In comparison with gRNA reads on day 1 before splitting cells into different culture conditions, there was a significant reduction of gRNA counts on day 5 under normoxia for the mitochondrial ribosomal genes (*Figure 4A and B*), in contrast to 1% oxygen which showed no difference between day 5 and day 1. Even on day 10, gRNA counts for mitochondrial translation genes were relatively more abundant in 1% oxygen than normoxia (p=0.085; *Figure 4B*), indicating that hypoxia buffers the blockade effect of mitochondrial protein translation. The large-scale CRISPR screening of 1095 cell lines under normoxia performed by the DepMAP project showed that the knockout effect of human mitochondrial ribosomal gene *MPRS22* (as one example) was correlated with *HIF1A* expression (*Figure 4C*), supporting the hypothesis that cells may survive longer under hypoxia when mitochondrial translation is inhibited. The electron transport chain is composed of five complexes (complex I-V), which drives ATP production by relaying electrons to oxygen. Surprisingly, the knockout of each complex gave rise to distinct fitness outcomes under normoxia and 1% oxygen tensions in monolayer and 3D culture. While knockout of complex I and IV led to a cell fitness advantage in all three conditions (which was opposite to K562 and U2OS cells which cannot tolerate complex I loss under normoxia *Thomas et al., 2021*; *Jain et al., 2020*), loss of function of complex II and III seemed to be detrimental to NEJF10 cells in both 21% and 1% oxygen in monolayer, although gRNA counts were relatively enriched in 3D (*Figure 4D*). However, genetic deletion of complex V, the ATP synthase, led to an adverse effect on cell survival under normoxia, opposite to 1% oxygen conditions in which cells gained a fitness advantage with knockout of complex V (*Figure 4D*). For example, gRNAs for

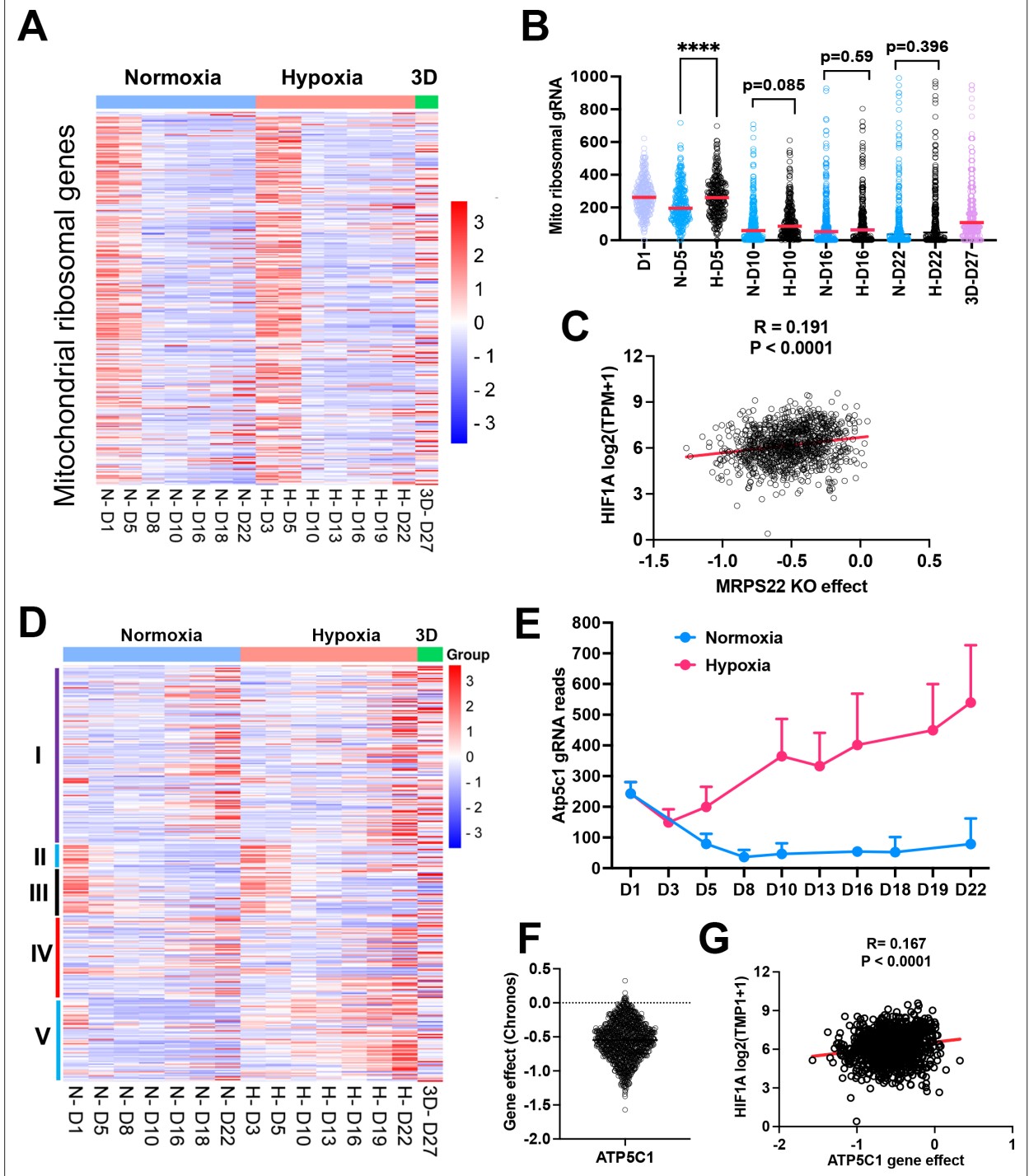

**Figure 4.** The effect of mitochondrial compartments on cell fitness. (**A**) Heatmap showing the gRNA reads for mitochondrial ribosomal genes at different time points in 2D normoxia, 2D hypoxia and 3D normoxia. Scale bar indicates Z score. (**B**) Comparison of gRNA reads for mitochondrial ribosomal genes at different time points in 2D normoxia, 2D hypoxia and 3D normoxia. ****p<0.0001. p value is calculated by student t test for the last time point data. Data = mean +/-SD. (**C**) Spearman correlation of *MRPS22* CRISPR KO effect vs *HIF-1A* gene expression in 1078 human cell lines. (**D**) Heatmap showing the gRNA reads for mitochondrial electron transport genes at different time points in 2D normoxia, 2D hypoxia, and 3D normoxia. Scale bar indicates Z socre. (**E**) The gRNA reads for *Atp5c1* at different time points in 2D normoxia and 2D hypoxia. Error bar data = mean +/-SD. (**F**) CRIPSR KO effect of *ATP5C1* in 1078 human cell lines. Data were extracted from DepMAP database (https://depmap.org/portal/home/). (**G**) Spearman correlation of *ATP5C1* CRISPR KO effect vs *HIF-1A* gene expression in 1078 human cell lines.

The online version of this article includes the following figure supplement(s) for figure 4:

**Figure supplement 1.** Spearman correlation analysis of IACS-10759 effect vs HIF1A expression.

the *Atp5c1* gene, encoding one key component of ATP synthase, were gradually depleted under 21% oxygen but increased under 1% oxygen over time (*Figure 4E*). The large-scale CRISPR screening of 1095 cell lines under normoxia by the DepMAP project showed that human *ATP5C1* is a pan-essential gene (*Figure 4F*) because nearly all cells cannot tolerate the loss of *ATP5C1*. There was a negative correlation between *ATPC1* knockout and *HIF1A* expression (*Figure 4G*). Because *HIF1A* can serve as a hypoxia surrogate marker, this negative correlation independently validated our observation that cells with loss of the ATPase survive better in hypoxia than normoxia.

Therapeutics targeting the oxidative phosphorylation (OXPHOS) pathway are being evaluated in clinical trials. In two recent phase one clinical trials for treatment of advanced solid cancers and acute myeloid leukemia, a complex I inhibitor IACS-010759 showed limited antitumor activity (*Yap et al., 2023*). While in-depth study is needed to understand the mechanism by which IACS-010759 failed in clinic, the anticancer activity of IACS-010759 seemed to be inversely correlated with *HIF1A* expression (*Figure 4—figure supplement 1*). Therefore, the anticancer activity of IACS-010759 might be greatly blunted by hypoxic conditions in a tumor. Considering cancer cells such as NEJF10 could gain a fitness advantage when complex I is genetically inhibited, pharmacological inhibition of complex I could even promote tumor growth under some specific settings.

## Epigenetics and organogenesis pathways function as important fitness mechanisms in 3D culture

There were fewer genes which were enriched in the CRISPR screen whose loss of function promoted cell fitness under the three conditions. Protein interaction network analysis further validated context-specific dependencies such as heme biosynthesis genes (*Alas1*, *Hmbs*) under 21% oxygen in 2D culture, and fatty acid synthesis and mitochondrial respiratory-chain complex (MITRAC) genes (*Cox18*, *Surf1*, *Tmem70*) under 1% oxygen in 2D culture (*Figure 5A and B*). Under 3D culture conditions, protein interaction networks were enriched with genes involved in epigenetics (*Kmt2d*, *Bcor*, *Arid2*, *Phf21b*, etc), TGFβ-SMAD signaling pathway (*Tgfbr1*, *Tgfbr2*, *Tgfbr3*, *Smad3*, *Smad4*, *Runx3*), Hippo signaling pathway (*Nf2*, *Lats1*, *Sav1*, *Frmd6*), and M6A methyltransferase complex (*Mettl3*, *Mettl14*, *Wtap*; *Figure 5C*). After carefully examining the enriched genes based on literature annotation, we found that the signaling pathways involved in multicellular organogenesis such as TGFβ, Wnt, Hedgehog, and Notch pathways were enriched, particularly in 3D culture (*Figure 5D*). This finding supports the theory that breakdown of communication and coordination between cells (which is required for multi-cellularity) leads to tumorigenesis and cancer progression (*Trigos et al., 2018*; *Trigos et al., 2017*; *Chen et al., 2015*). TGFβ-SMAD2/3/4 pathway is known to be involved in tumorigenesis. Interestingly, the BMP-SMAD1/5/8 signaling was not enriched in the screenings, indicating the TGFβ-SMAD2/3/4 pathway is particularly important for restricting 3D cell proliferation.

Considering that the aforementioned organogenesis signaling pathways converge on nuclear gene transcription, the above-mentioned epigenetic modifiers are highly likely to be integrated into these signaling pathways and modulate gene transcription. Notably, most of these genes (*Kmt2d*, *Bcor*, *Smad*, *Nf2*, *Lats1*) are tumor suppressors in human cancers. KMT2D (also known as MLL4) is a methyltransferase that methylates H3K4, and which frequently exhibits loss of function mutations in a variety of human cancers (*Rao and Dou, 2015*; *Figure 5—figure supplement 1A*). Interestingly, DepMAP CRISPR screens showed that *KMT2D* was a pan-essential gene in 2D normoxia regard-less of its mutation status (*Figure 5—figure supplement 1B*). This was consistent with our results which showed *Kmt2d* gRNA depletion under both 21% and 1% oxygen tensions in 2D culture, and which contrasts with conventional understanding that cells should gain a proliferation advantage after knockout of a tumor suppressor gene. However, under 3D culture, *Kmt2d* gRNAs were remark-ably enriched (*Figure 5E*), which supports the tumor suppressor role of *KMT2D*. *BCOR* is another frequently mutated tumor suppressor gene (*Astolfi et al., 2019*). Knockout of *Bcor* in NEJF10 cells promoted cell fitness in hypoxia and more dramatically in 3D but had minimal effect in normoxia (*Figure 5E*). DepMAP data showed that the average effect of *BCOR* gene knockout was close to net zero (*Figure 5—figure supplement 2A*). These data underscore that careful interpretation of a CRISPR knockout phenotype for tumor suppressor genes needs to consider the cellular context. *Wtap*, *Mettl3,* and *Mettl14* encode proteins forming the WMM complex that acts as a N6-methyl-transferase to methylate adenosine residues at the N(6) position of some mRNAs (M6A) to regu-late mRNA stability. Similar to the fitness outcomes of *Kmt2d* in the three conditions, knockout of

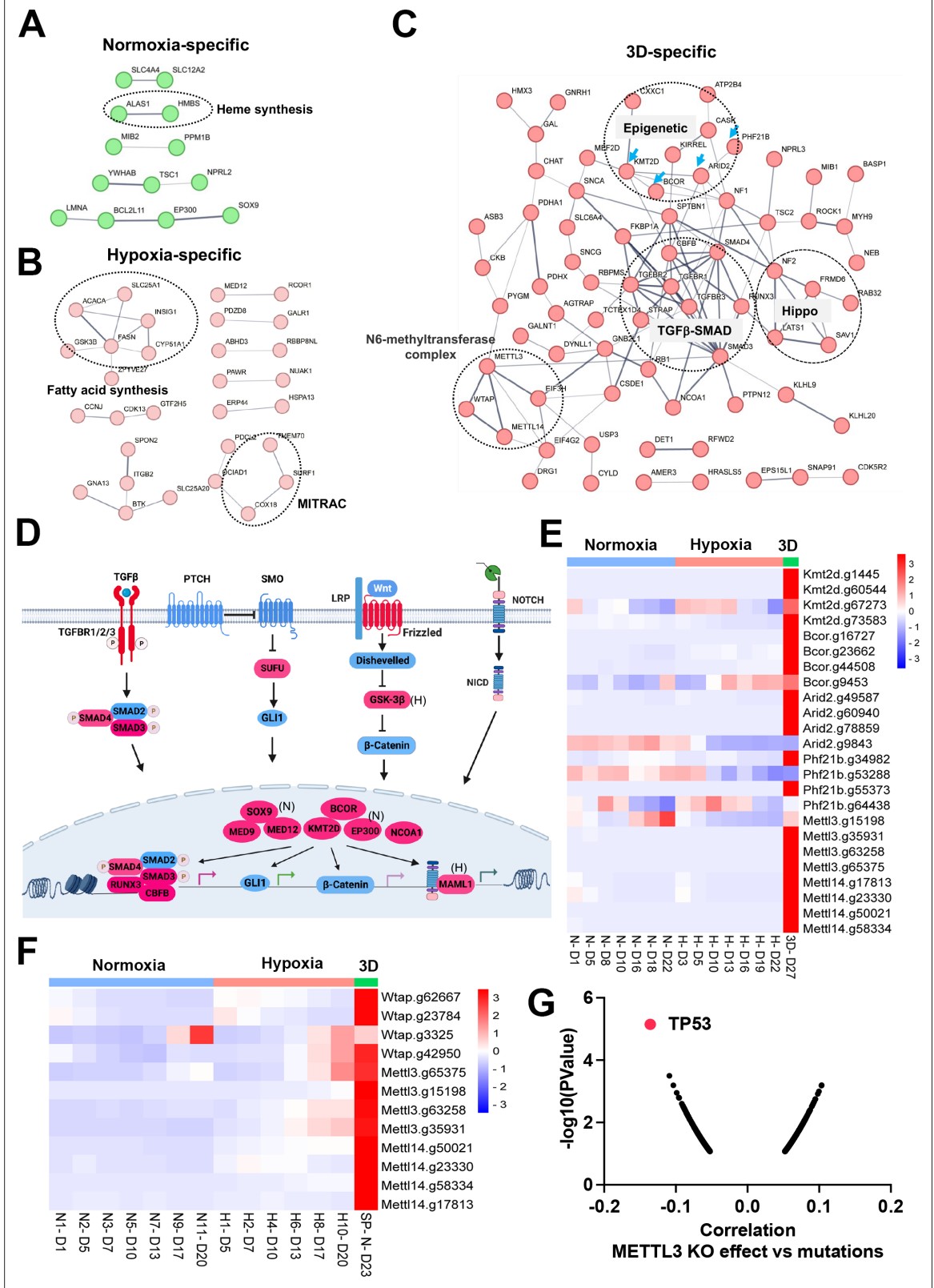

**Figure 5.** Organogenesis signaling and epigenetic modifiers constrain 3D cell proliferation. (**A–C**) Positive selection of pathways within protein-protein interaction networks enriched specifically in 2D normoxia, 2D hypoxia, and 3D normoxia. (**D**) The graphic illustrates organogenesis signaling pathways positively selected in 3D. Please note that we included hits from normoxia (Sox9, Ep300) and hypoxia (Maml1 and Gsk-3b) in the related pathways. Created with BioRender. (**E**) The heatmap for the normalized gRNA reads for epigenetic genes in 2D hypoxia and normoxia and 3D normoxia. Red color

*Figure 5 continued on next page*

Figure 5 continued

indicates positive selection. Scale bar indicates Z score. (F) The heatmap for the normalized gRNA reads for WMM complex of NEJF6 cells in 2D hypoxia and normoxia and 3D normoxia. Scale bar indicates Z score. (G) Spearman correlation of METTL3 CRISPR KO effect vs TP53 mutations in human cancer cells lines by analyzing DepMAP data (https://depmap.org/portal/home/).

The online version of this article includes the following figure supplement(s) for figure 5:

**Figure supplement 1.** Mutations in *KMT2D* in human cancers and its knockout effect in human cancer cell lines.

**Figure supplement 2.** Gene knockout effect in human cancer cell lines under 2D normoxia.

*Wtap*, *Mettl3* and *Mettl14* led to great gRNA count enrichment in 3D culture but reduction in 2D culture (*Figure 5E*). The 3D-specific role of the WMM complex was further verified in NEJF6 cells in which gRNA reads of WMM complex were remarkably increased under 3D culture as in NEJF10 cells (*Figure 5F*, *Supplementary file 10*). DepMAP data showed that *WTAP*, *METTL3,* and *METTL14* are commonly essential to most if not all cells (*Figure 5—figure supplement 2A*). While most previous studies recognize that METTL3 and METTL14 are important to cancer cell survival, one study showed that METTL3 enhanced the tumor suppression activity of p53 (*Raj et al., 2022*). A recent study reported that knockout of *Mettl3* enhances liver tumorigenesis in multiple mouse models (*Wei et al., 2023*). Two studies also indicate that METTL14 acts as a tumor suppressor by facilitating DNA repair or modulating glycolysis (*Hou et al., 2023*; *Yang et al., 2021*). By analyzing DepMAP data, we further confirmed the functional connection of the WMM complex with p53. The knockout effect of *METTL3* was negatively correlated with the presence of *TP53* mutations (*Figure 5G*), positively correlated with the knockout effect of *TP53*, but negatively correlated with the knockout of *MDM2* (*Figure 5—figure supplement 2B, C*). In line with these observations, the knockout effect of *METTL3* was positively correlated with *MDM2* expression (*Figure 5—figure supplement 2D*). MDM2 is known to antagonize the functions of p53. Thus, despite the overall detrimental effect of WMM knockout under 2D normoxia conditions from the DepMAP data, we were able to establish the genetic connection of WMM-p53-MDM2, supporting the observation WMM is tumor suppressive. It is important to note that we cannot exclude the possibility that these genes may have cancer-specific functions, and may behave differently in different cancer lineages. Taken together, our data indicate that organogenesis signaling and epigenetic regulators could behave drastically differently in 2D vs 3D culture conditions.

## Context-specific fitness genes with altered gene expression in hypoxia and 3D are enriched with metabolic pathways

Two previous studies showed that HIF targets were not enriched in identified fitness genes under hypoxia (*Thomas et al., 2021*; *Jain et al., 2020*). HIF mainly upregulates gene expression and the downregulated genes caused by hypoxia were not directly induced by HIF (*Lombardi et al., 2022*). In the current study, pathway enrichment analysis showed that the hypoxia-downregulated genes were negatively selected by CRISPR knockout (*Figure 6—figure supplement 1A*). These negative fitness genes were enriched in the pathways of MYC, Wnt-β-Catenin and NOTCH4, as well as mitochondrial metabolism (*Figure 6—figure supplement 1A*). However, among the positive selection fitness genes whose expression was altered by hypoxia, lipid metabolism genes were most significantly enriched (*Figure 6—figure supplement 1B*). Among the genes whose expression was altered by 3D culture, the hypoxia-downregulated genes were also negatively selected, probably due to induction of the hypoxia pathway by 3D culture (*Figure 6—figure supplement 1C*). Different from hypoxia, however, the 3D genes in negative selection were enriched in pathways of MYCN, BMP2, DREAM complex and genetic modifiers such as DNA and histone methylases, and histone acetyltransferases (*Figure 6—figure supplement 1C*). For the 3D genes positively selected in the screening, the TGFβ-SMAD pathway and NOTCH pathway were remarkably enriched (*Figure 6—figure supplement 1D*), further suggesting that loss of genes in multicellular communication promotes tumor progression.

Next, we examined context-specific fitness genes whose expression and CRISPR gRNAs were selectively altered in 1% oxygen or 3D conditions. We only found 13 and 14 genes in negative and positive selection screenings respectively under 1% oxygen, 20 and 20 genes in negative and positive selection screenings respectively in 3D culture (*Figure 6A and B*, *Supplementary file 11*). Several hypoxia-inducible genes involved in mitochondrial import (*Cox17* and *Tomm20*) and mitochondrial translation (*Mrps34*, *Mrpl54*) were selectively essential to cell fitness under 1%

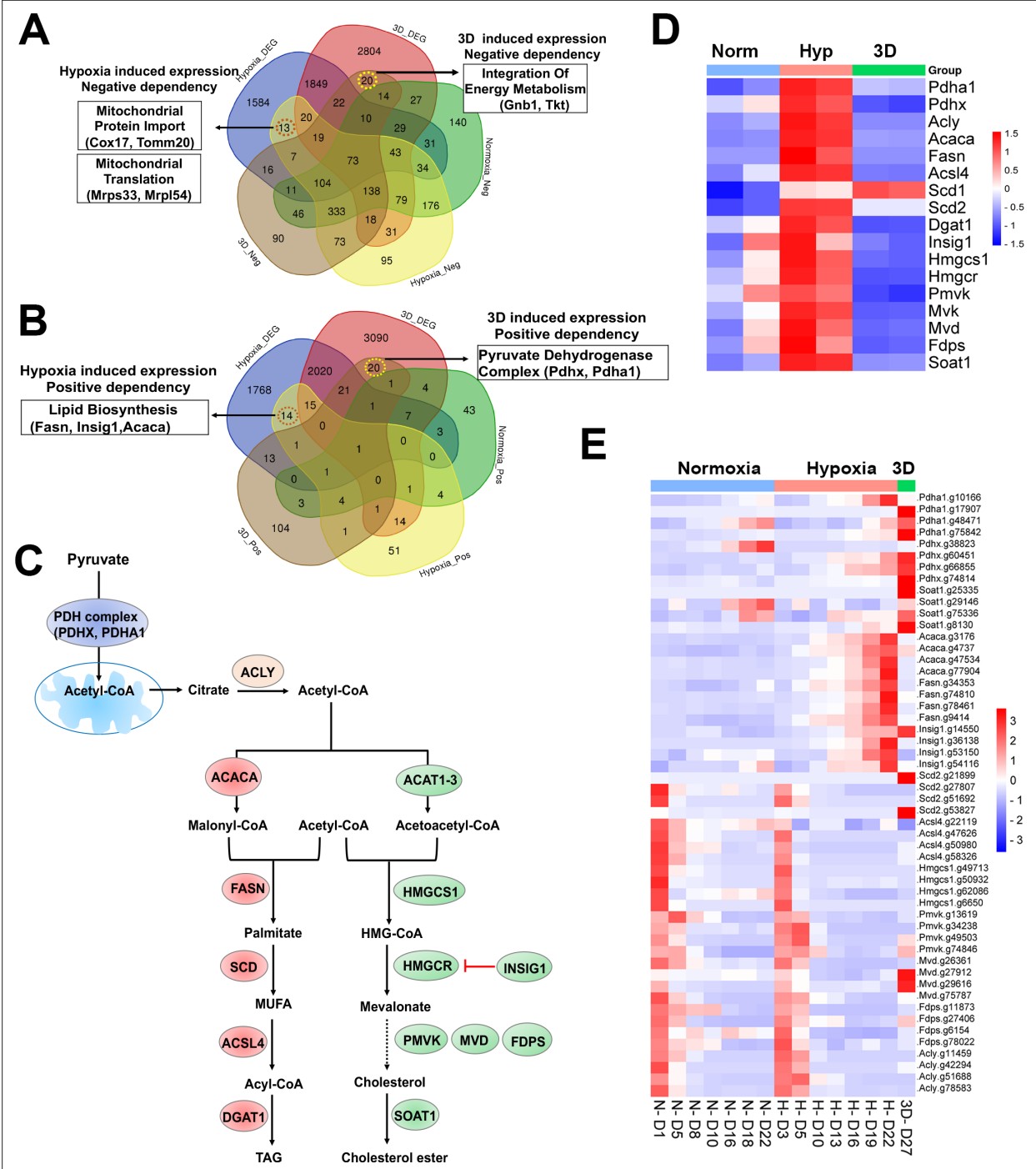

**Figure 6.** Selective fitness of lipid metabolisms. (**A**) Venn diagram showing 13 and 20 genes with negative dependency in specifically hypoxia and 3D induced conditions, respectively, are selectively induced in hypoxia or 3D. (**B**) Venn diagram showing 14 and 20 genes with positive dependency in specifically hypoxia and 3D induced conditions, respectively, are selectively induced in hypoxia or 3D. (**C**) A diagram showing lipid biosynthesis pathway of unsaturated fatty acids and cholesterol. (**D**) The heatmap showing the gene expression involved in lipid biosynthesis. Scale bar indicates Z score. (**E**) The heatmap showing the normalized gRNA reads for genes involved in lipid biosynthesis in 2D normoxia and hypoxia, 3D normoxia. Scale bar indicates Z score.

The online version of this article includes the following figure supplement(s) for figure 6:

**Figure supplement 1.** Pathways enrichment for fitness genes whose expression is altered in hypoxia and 3D.

**Figure supplement 2.** Inhibition effect of lipid metabolic genes of human homologs.

oxygen tension (**Figure 6A**). *Pdgfrb*, another hypoxia-inducible gene involved in cell proliferation and angiogenesis, was also an important fitness gene in 1% oxygen. However, hypoxia-inducible genes involved in regulation of fatty acids and cholesterol (*Fasn*, *Insig1*, *Acaca*) were positively selected under 1% oxygen tension (**Figure 6B–E**). ACACA and FASN are responsible for the de novo biosynthesis of long-chain saturated fatty acids starting from acetyl-CoA and malonyl-CoA in the presence of NADPH, while INSIG1 negatively regulates HMGCR for inhibition of cholesterol synthesis (**Figure 6C**). The selective 3D inducible genes involved in integration of energy metabolism (*Gnb1*, *Tkt*) were particularly essential to cells in 3D culture (**Figure 6A**), while those involved in regulation of the pyruvate dehydrogenase (PDH) complex (*Pdhx*, *Pdha1*) were positively selected after knockout in 3D culture albeit to a lesser degree under 1% oxygen (**Figure 6B and E**). GNB1 is a guanine nucleotide-binding protein (G protein) involved as a transducer in trans-membrane signaling, and whose gain of function mutations promote myeloid transformation (**Yoda et al., 2015**). TKT is a thiamine-dependent enzyme which plays a role in the pentose phosphate pathway. The PDH complex is known to convert pyruvate to acetyl-CoA for citrate synthesis in the tricarboxylic acid cycle and cholesterol and fatty acid synthesis (**Figure 6C**). It is likely that blockade of acetyl-CoA production by PDH knockout may force cells to use alternative energy sources under hypoxic and 3D conditions, averting the Warburg effect and promoting cell survival under limited oxygen and nutrient availability in 3D spheroids. The CRISPR results were corroborated by the findings that the genes involved in regulation of PDH complex were selectively upregulated in normoxic 3D conditions (**Figure 1A**). This hypothesis awaits further validation in future studies. It is noteworthy that the activity of PDH is regulated by pyruvate dehydrogenase kinase (PDK). *Pdk1* underwent distinct alternative splicing changes in hypoxia and 3D (**Figure 1—figure supplement 1E**), which may consequently affect the activity of PDH.

## Distinct dependency of fatty acid and cholesterol synthesis pathways

Previous studies identified lipid metabolic genes which are critical to cell fitness under hypoxia (**Jain et al., 2020**). However, lipid metabolic genes showed a more complex fitness phenotype, although their expression was induced by hypoxia (**Figure 6D**). For example, blockade of saturated fatty acid synthesis by knockout of *Fasn* and *Acaca* promoted cell fitness under hypoxia, while *Scd2*, encoding a stearoyl-CoA desaturase that utilizes oxygen and electrons from reduced cytochrome b5 to introduce the first double bond into saturated fatty acyl-CoA substrates, was essential to NEJF10 cells in both normoxia and hypoxia (**Figure 6E**). This was distinct from the K562 cells which were selectively sensitive to *SCD* (human homolog of *Scd2*) KO in hypoxia (**Jain et al., 2020**). *Acsl4*, encoding acyl-CoA synthetase long chain family member 4 protein that was selectively essential in hypoxia to K562 cells (**Jain et al., 2020**), was essential to NEJF10 cells in all culture conditions (**Figure 6E**). While the peroxisome was reported to be critical to K562 cell survival grown in hypoxia (**Jain et al., 2020**), NEJF10 cells were not as sensitive to peroxisome loss. Analysis of DepMAP data revealed that the knockout effect of peroxisome genes (i.e. *PEX1*) had no significant correlation with the expression of either *HIF1A* or *MYC* while *SCD* knockout effect was negatively correlated with *MYC* expression (**Figure 6—figure supplement 2A–D**), suggesting that *SCD* is an MYC cancer dependency gene. However, *SCD* knockout effect was positively correlated with *HIF1A* expression. Similar to *SCD*, *ACSL4* knockout effect was also negatively correlated with *MYC* expression but positively correlated with *HIF1A* expression (**Figure 6—figure supplement 2E, F**). The effect of the SCD inhibitors A-939572 and MK-8245 was negatively impacted by high *ACSL4* expression (**Figure 6—figure supplement 2G, H**), in line with the fact that ACSL4 is downstream of SCD (**Figure 6C**). These data further support that SCD and ACSL4 are posited in one metabolic pathway in regulating cell fitness. Taken together, our data suggest that saturated and non-saturated fatty acid synthesis exert opposite functions in regulating cell fitness, at least in NEJF10 cells. The ratio of saturated vs unsaturated fatty acids is critical to cell survival (**Ackerman and Simon, 2014**). This possibly because saturated fatty acids are toxic to cells due to inducing endoplasmic reticulum (ER) stress (**Ackerman and Simon, 2014**). Hypoxia induces high expression of *Acaca* and *Fasn* in NEJF10 cells indicating that hypoxia promotes saturated fatty acid synthesis, which is in line with the observation by Jain (**Han et al., 2020**). The beneficial effect of *Fasn* and *Acaca* KO to NEJF10 under hypoxia is probably due to reduction of saturated fatty acid synthesis, and this hypothesis needs to be tested in the future. Although *Scd2* in NEJF10 cells was induced by hypoxia, it is possible that this is a compensatory induction because SCD proteins need

oxygen for their activity. Thus, cells may be particularly sensitive to inhibition of stearoyl-CoA desaturase under hypoxic conditions.

The cholesterol biosynthesis pathway, another downstream metabolic branch of acetyl-CoA (*Figure 6C*), was essential to cell survival since under either culture condition cells cannot survive after knockout of *Hmgcs1*, which encodes 3-hydroxy-3-methylglutaryl-CoA synthase 1, an enzyme catalyzing the condensation of acetyl-CoA with acetoacetyl-CoA to form HMG-CoA for cholesterol synthesis. The downstream enzymes for cholesterol synthesis such as *Pmvk*, *Mvd*, *Fdps* were essential for cell fitness under both normoxia and hypoxia in monolayer culture (*Figure 6E*). Considering that cholesterol is required for membrane biogenesis and maintains the integrity and fluidity of cell membranes, cancer cells may not survive when the cholesterol synthesis pathway is fully shut down. MYC-driven cancers may be particularly sensitive to interruption of cholesterol synthesis since MYC is linked to dysregulation of cholesterol transport and storage (*Hall et al., 2020*). A previous study has shown that inhibition of cholesterol synthesis by statins prevents and reverses MYC-induced lymphomagenesis (*Shachaf et al., 2007*). Therefore, targeting cholesterol synthesis might be an option for MYC-driven cancers.

## Synthetic lethality of partial loss of PRMT5 under 3D but not 2D culture

The spliceosome is essential to cell survival regardless of culture conditions (*Figure 2—figure supplement 1C*). One of these key essential genes is *Prmt5* (*Figure 7A*), which encodes splicing factor PRMT5. PRMT5 is required for survival of MYC-driven cancer cells (*Koh et al., 2015*), and has been extensively studied as a potential cancer therapeutic target. The essentiality of PRMT5 was further validated in another independent CRISPR screen in the NEJF6 cell line (*Supplementary file 10*), another MYC-driven liver cancer cell line derived from ABC-Myc mouse liver tumor (*Fang et al., 2023*). Interestingly, shRNA-mediated knockdown of *Prmt5* significantly reduced the growth of NEJF10 spheroids while minimal effect was seen when NEJF10 was cultured in monolayer under 21% and 1% oxygen tension (*Figure 7B–E*). As shRNAs usually partially deplete the expression of target genes as evidenced by the western blot analysis, these data indicate that complete loss of *Prmt5* is needed to inhibit NEJF10 proliferation under 2D culture conditions. We further validated the role of *Prmt5* by genetically deleting *Prmt5* from liver via breeding a floxed *Prmt5* mouse strain with the ABC-Myc mice (*Fang et al., 2023*). In comparison with the ABC-Myc control mice that rapidly developed liver tumors, knockout of either one allele or both alleles of *Prmt5* extended comparable mouse survival (*Figure 7F*). H&E staining of the liver tissues revealed that deletion of even one *Prmt5* allele led to massive necrosis in some tumor regions with inflammatory cell infiltration (as evidenced by macrophage surrounding of the necrotic areas). This was not observed in the control mice (*Figure 7G*, *Figure 7—figure supplement 1*). These data verified the importance of *Prmt5* for tumor cell survival under multicellular settings, and demonstrate even partial loss of *Prmt5* function in vivo may lead to a comparable effect to complete *Prmt5* loss. Notably, the control mouse livers with *Prtm5* knockout appeared to be normal, indicating that *Prtm5* is essential for the MYC-transformed cancer cells but not for the non-transformed liver cells.

## Epigenetic reprogramming in 3D culture leads to downregulation of *Mtap*

We sought to understand why partial loss of *Prmt5* affected cell proliferation in 3D but not in 2D culture. It is well known that the *MTAP* (encoding 5-methylthioadenosine phosphorylase) deletion is synthetically lethal to genetic ablation of *PRMT5* (*Kryukov et al., 2016*), and this finding has been translated to clinical trials by using PRMT5 inhibitors for tumors with *MTAP* deletions. Here, we further verified the expression of *MTAP* and *PRMT5* knockdown effect (*Figure 7—figure supplement 2*). Since we have observed unique pre-mRNA splicing changes in 3D culture, we examined if *Mtap* underwent distinct splicing. Indeed, in comparison with the 2D culture which showed a similar splicing pattern in normoxia and hypoxia, *Mtap* displayed distinct exon skipping of exon 2 and/or 3 in 3D culture (*Figure 7—figure supplement 3*), which may potentially hinder its enzymatic activity by abrogating sulfate/phosphate binding (*Appleby et al., 1999*). However, only a small fraction of *Mtap* pre-mRNA transcripts underwent these events. Notably, the exon junction reads of whole *Mtap* pre-mRNA transcript were greatly reduced in 3D culture, consistent with the reduction of *Mtap* transcription in 3D vs 2D culture conditions (*Figure 7H*). We further verified this by quantitative real-time

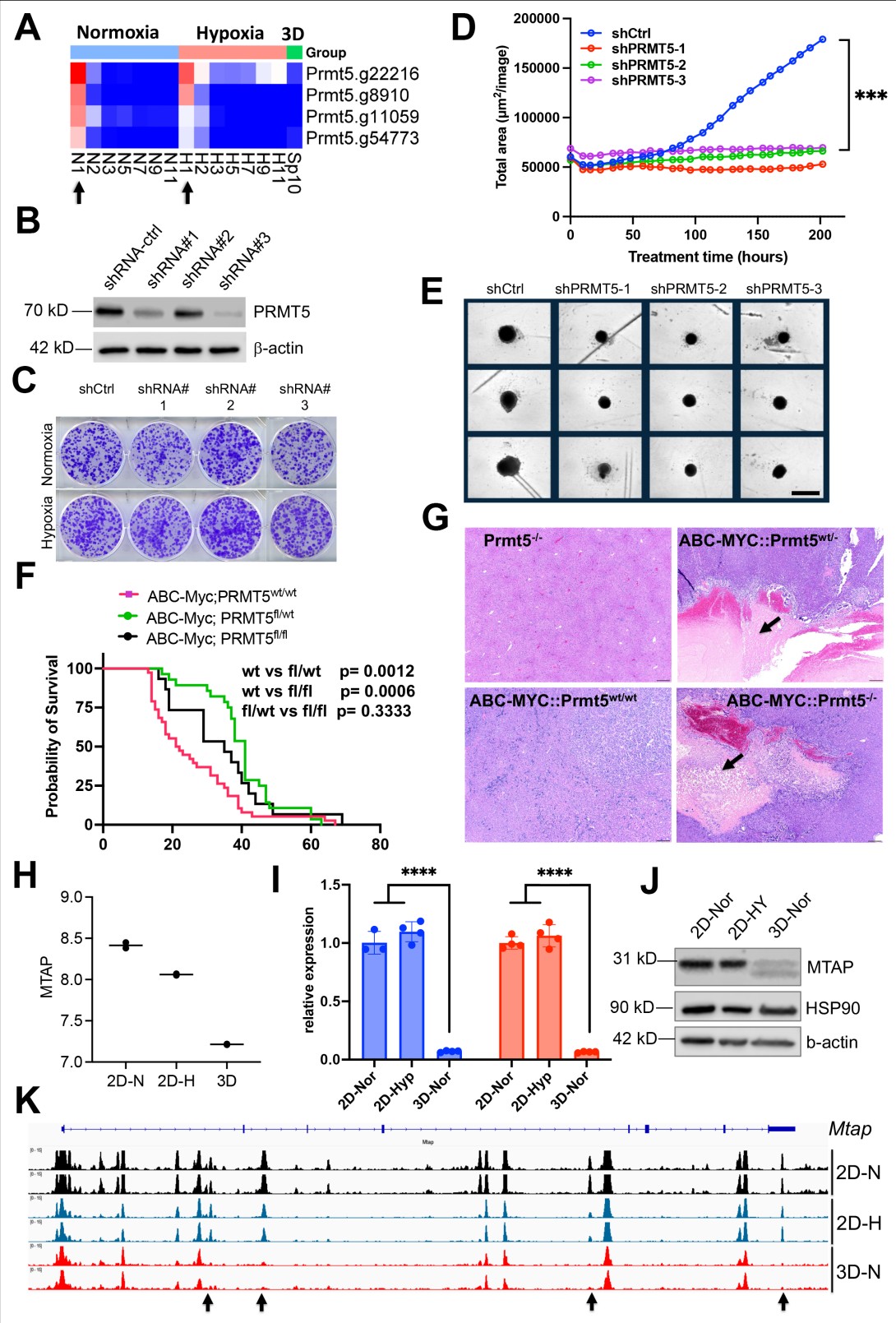

**Figure 7.** Synthetic lethality of partial Prtm5 loss with 3D. (**A**) The heatmap showing the normalized gRNA reads for *Prmt5* in 2D normoxia and hypoxia, 3D normoxia. Arrows indicate the first samples harvested for analysis. (**B**) Western blot analysis with indicated antibodies after *Prtm5* knockdown in NEJF10 cells. (**C**) Colony formation of NEJF10 cells after 5 days of *Prmt5* knockdown in 2D normoxia and hypoxia. Cells were stained with crystal violet. (**D**) Incucyte monitoring of cell proliferation grown in 3D after *Prtm5* knockdown in NEJF10 cells. ***p<0.001 for comparison of shCtrl with each of

*Figure 7 continued on next page*

*Figure 7 continued*

shPrmt5. p value is calculated by student t test by comparing the last reading. N=3 per group. Data = mean +/- SD. (**E**) Snapshot of NEJF10 cell with or without *Prmt5* knockdown in 3D. Scale bar = 400μm. (**F**) Kaplan-Meier survival for mice of *ABC-Myc::Prmt5+/+*, *ABC-Myc::Prmt5+/-*, and *ABC-Myc::Prmt5-/-*. p values are calculated by log-rank test. (**G**) Hematoxylin and eosin stain for tumor tissues obtained from mice of *Prtm5-/-*, *ABC-Myc::Prmt5+/+*, *ABC-Myc::Prmt5+/-, and ABC-Myc::Prmt5-/-*. Arrows indicate necrosis in tumor areas. Scale bar = 500 μm. (**H**) Quantification of normalized *Mtap* mRNA under 2D normoxia, 2D hypoxia, and 3D normoxia from RNA-seq results. n=2 for each group. (**I**) Real-time quantitative PCR to determine the expression of *Mtap* in NEJF10 cells cultured under 3D normoxia, 2D normoxia, and hypoxia for 3 days. Blue and orange indicate results obtained from two different pairs of PCR primers against *Mtap*. n=3 for each group.****p<0.0001 student t test. (**J**) Western blot analysis with indicated antibodies of NEJF10 whole cell lysates cultured from 3D normoxia, 2D normoxia, and hypoxia for 3 days. (**K**) IGV snapshot showing the ATAC-seq result for *Mtap* gene. The black arrow indicates the enhancers in *Mtap* gene are selectively lost in 3D.

The online version of this article includes the following source data and figure supplement(s) for figure 7:

**Source data 1.** The source data contains original raw data of western blots for ***Figure 7B***, labelled.

**Source data 2.** The source data contains original raw data of western blots for ***Figure 7B***.

**Source data 3.** The source data contains original raw data of western blots for ***Figure 7J***, labelled.

**Source data 4.** The source data contains original raw data of western blots for ***Figure 7J***.

**Figure supplement 1.** Necrosis and inflammation of liver tumor with Prmt5 knockout.

**Figure supplement 2.** PRMT5 knockdown effect vs MTAP expression.

**Figure supplement 3.** Splicing events of *Mtap* gene under different culture conditions.

PCR (***Figure 7I***) and western blot analysis (***Figure 7J***). To determine the mechanism by which *Mtap* was reduced in 3D culture, we examined the chromatin accessibility at the *Mtap* genomic locus. The results showed a drastic effect of culture conditions on *Mtap* chromatin accessibility in which four open chromatin regions were closed in 3D culture (***Figure 7K***). These data suggest that the enhancer activity driving *Mtap* expression was repressed, leading to the downregulation of *Mtap* expression, and consequently, synthetic lethality with partial loss of Prmt5.

## Discussion

The heterogeneity of an organ or tumor at single cell level is not only determined by genetic and epigenetic factors but also by its surrounding microenvironment such as nutrient and oxygen availability and cell-cell interaction. We do not entirely understand how each cell adapts to its endogenous and exogenous milieus for cellular fitness. While technologies such as scRNA-seq and spatial transcriptomics have helped to understand cellular heterogeneity, it is challenging to dissect each cell's fate in an in vivo setting like an organ or a tumor mass. Genome-wide CRISPR screening approach in combination with certain environmental conditions in an in vitro setting are valuable for understanding the combined genetic and environmental interactions that determine cell fitness. In this study, we used an MYC-driven murine liver cancer model (***Fang et al., 2023***), and successfully identified context-specific fitness genes and pathways, as well as commonly shared fitness genes, in monolayer culture under 21% and 1% oxygen tensions and 3D spheroid culture under 21% oxygen tension. Notably, our study revealed that (1) organogenesis pathways such as TGFβ-SMAD are enriched in 3D spheroids under positive selection; (2) epigenetic modifier genes encoding BCOR, KMT2D, METTL3 and METTL14 act in different ways in 2D vs. 3D culture; (3) Loss of the VHL-HIF1 pathway is incompatible with cell survival in normoxic 2D conditions, but not hypoxic 2D or normoxic 3D conditions; (4) Distinct requirements for each complex of the electron transport chain exist in normoxia, hypoxia and 3D; (5) Distinct requirements for fatty acid and cholesterol synthesis pathways exist in normoxia and hypoxia and 3D; and (6) Epigenetic reprogramming of *Mtap* in 3D culture leads to context-dependent synthetic lethality to *Prmt5* knockdown. Overall, our studies demonstrated that cancer cells have distinct fitness mechanisms which are dependent on culture conditions. While these findings may not be overtly surprising, our study also revealed nuanced, counterintuitive findings such as each component of the same signaling pathway (i.e. complex of oxidative phosphorylation) exhibiting distinct effects on cell fitness when genetically deleted. Nevertheless, the question of why knockout of each complex of the electron chain reaction gave rise to different fitness outcomes under different cellular context remains to be answered.

Epigenetic modifiers such as KMT2D, BCOR, METTL3, and METTL14 were a limiting factor for uncontrolled cell proliferation in 3D spheroids, which may reveal the mechanism of how they function as tumor suppressors in human cancer. These epigenetic modifiers likely maintain the cellular homeostasis during organogenesis, and when disrupted, tumorigenesis ensues. Surprisingly, knockout of *Kmt2d* and *Mettl3/Mettl14* led to fitness defects in 2D culture under 21% oxygen tension. The DepMAP data also showed that cells cannot survive when *KMT2D* is deleted regardless of its mutation status. We speculate that cell-cell communication in 3D culture more closely represents organogenesis in vivo, which needs to be well controlled to prevent aberrant growth. While under a 2D setting, such cell-cell communications do not exist, alternative signaling acts and gives rise to different phenotypes from 3D culture. While the mechanisms accounting for the phenotypic discrepancies in 2D vs 3D conditions for these epigenetic modifiers await elucidation in future studies, our current findings demonstrate that caution should be taken when interpreting the phenotypic screening of these epigenetic modifiers under conventional cell culture conditions.

While HIF pathways were induced by both 1% oxygen and 3D culture, CRISPR screening showed that fewer HIF targets constitute fitness genes in this model system. Instead, hypoxia-inducible genes that were not HIF targets were selectively essential at different oxygen tensions. These results were consistent with previous studies (*Thomas et al., 2021*; *Jain et al., 2020*). Nevertheless, we found that *Vhl* and *Hif1a* were essential for cell survival in NEJF10 cells under 21% and 1% oxygen respectively, and this was not observed in K562 and U2OS cells (*Thomas et al., 2021*; *Jain et al., 2020*), suggesting a cell type-specific effect. The previous study found that lipid metabolism and peroxisome genes in K562 cells are essential in hypoxia (*Jain et al., 2020*). However, the peroxisome pathway was not enriched in the MYC-driven NEJF10 cells in either culture condition, suggesting that peroxisome-mediated lipid metabolism might not be essential to MYC-driven cancer. Interestingly, genes responsible for saturated (*Fasn*, *Acaca*) and non-saturated fatty acid synthesis (*Scd2*) or fatty acid catabolism (*Acsl4*) exert opposite functions in cell fitness. Considering that both MYC and HIF play critical roles in regulating metabolic gene expression, further dissection of their interaction under different cellular contexts may help us understand the context-specific fitness genes.

# Materials and methods

## Key resources table

| Reagent type (species) or resource | Designation | Source or reference | Identifiers | Additional information |
|---|---|---|---|---|
| Strain, strain background (Mouse, B6, both male and female) | Albumin-Cre | THE JACKSON LABORATORY | RRID:IMSR_JAX:003574 | |
| Strain, strain background (Mouse, B6, both male and female) | Rosa26StopFL-MYC | THE JACKSON LABORATORY | RRID:IMSR_JAX:020458 | |
| Strain, strain background (Mouse, B6/129 mix both male and female) | Prmt5 flox | THE JACKSON LABORATORY | RRID:IMSR_JAX:034414 | |
| Cell line (*Murine*) | NEJF10 | This paper | N/A | Cell line maintained in Jun Yang lab |
| Cell line (*Murine*) | NEJF6 | This paper | N/A | Cell line maintained in Jun Yang lab |
| Cell line (*Murine*) | NEJF10-shCtrl | This paper | N/A | Cell line maintained in Jun Yang lab |

*Continued on next page*

*Continued*

| Reagent type (species) or resource | Designation | Source or reference | Identifiers | Additional information |
|---|---|---|---|---|
| Cell line (*Murine*) | NEJF10-shPRMT5-1 | This paper | N/A | Cell line maintained in Jun Yang lab |
| Cell line (*Murine*) | NEJF10-shPRMT5-2 | This paper | N/A | Cell line maintained in Jun Yang lab |
| Cell line (*Murine*) | NEJF10-shPRMT5-3 | This paper | N/A | Cell line maintained in Jun Yang lab |
| Cell line (*Homo sapiens*) | U2OS | ATCC | HTB96 | |
| Cell line (*Homo sapiens*) | HCT116 | ATCC | CCL-247 | |
| Cell line (*Homo sapiens*) | HepG2 | ATCC | HB-8065 | |
| Cell line (*Homo sapiens*) | HepG2-LeveV2 | This paper | N/A | Cell line maintained in Jun Yang lab |
| Cell line (*Homo sapiens*) | HepG2-VHL-KO-1 | This paper | N/A | Cell line maintained in Jun Yang lab |
| Cell line (*Homo sapiens*) | HepG2-VHL-KO-2 | This paper | N/A | Cell line maintained in Jun Yang lab |
| Antibody | Anti-HIF1α (Rabbit, polyclonal) | Cayman | Cat# 10006421, RRID:AB_409037 | WB (1:200) |
| Antibody | Anti-HIF2α (Rabbit, polyclonal) | Novus | Cat# NB100-122, RRID:AB_10002593 | WB (1:500) |
| Antibody | Anti-VHL (Rabbit, polyclonal) | Cell Signaling Technology | Cat# 68547 S, RRID:AB_2716279 | WB (1:1000) |
| Antibody | Anti-PRMT5 (Rabbit, polyclonal) | ABclonal | Cat# A1520, RRID:AB_2762092 | WB (1:1000) |
| Antibody | Anti-MTAP (Rabbit, polyclonal) | Cell Signaling Technology | Cat# 4158 S, RRID:AB_1904054 | WB (1:1000) |
| Antibody | Anti-HSP90 (Mouse, monoclonal) | Santa Cruz | Cat# sc13119, RRID:AB_675659 | WB (1:1000) |
| Antibody | Anti-ACTIN (Mouse, monoclonal) | Sigma | Cat# A1978, RRID:AB_4766 | WB (1:5000) |
| Antibody | Anti-MAC2 (Rat, monoclonal) | Accurate/CEDARLANE | Cat# CL-8942AP, RRID:AB_10060357 | IHC (1:1000) |
| Recombinant DNA reagent | LeveV2-vector | Addgene | 52961 | |
| Recombinant DNA reagent | VHL-gRNA-1 | GenScript | SC1678 | |
| Recombinant DNA reagent | VHL-gRNA-2 | GenScript | SC1678 | |
| Recombinant DNA reagent | shCtrl | VectorBuilder | VB010000-0009mxc | |
| Recombinant DNA reagent | shPRMT5-1 | VectorBuilder | VB900070-3948cva | |
| Recombinant DNA reagent | shPRMT5-2 | VectorBuilder | VB900070-3942srh | |

*Continued*

| Reagent type (species) or resource | Designation | Source or reference | Identifiers | Additional information |
|---|---|---|---|---|
| Recombinant DNA reagent | shPRMT5-3 | VectorBuilder | VB900080-7913zan | |
| Sequence-based reagent | AlbCre genotyping primer1 | Integrated DNA Technologies | 5'-TGCAAACATCACATGCACAC-3' | |
| Sequence-based reagent | AlbCre genotyping primer2 | Integrated DNA Technologies | 5'-GAAGCAGAAGCTTAGGAAGAT GG –3' | |
| Sequence-based reagent | AlbCre genotyping primer3 | Integrated DNA Technologies | 5'-TTGGCCCCTTACCATAACTG –3' | |
| Sequence-based reagent | Rosa26StopFLMYC WT genotyping primer1 | Integrated DNA Technologies | 5'-CCAAAGTCGCTCTGAGTTGTTATC-3' | |
| Sequence-based reagent | Rosa26StopFLMYC genotyping primer2 WT | Integrated DNA Technologies | 5'- GAGCGGGAGAAATGGATATG –3' | |
| Sequence-based reagent | Rosa26StopFLMYC MYC genotyping primer1 | Integrated DNA Technologies | 5'- CCAAGAGGGTCAAGTTGGA –3' | |
| Sequence-based reagent | Rosa26StopFLMYC MYC genotyping primer2 | Integrated DNA Technologies | 5'-GCAATATGGTGGAAAATAAC-3' | |
| Sequence-based reagent | PRMT5 flox genotyping primer1 | Integrated DNA Technologies | 5'- GATACCAGGACCCACACTGG-3' | |
| Sequence-based reagent | PRMT5 flox genotyping primer2 | Integrated DNA Technologies | 5'-CTTAGGGGCTTCATGCATCT-3' | |
| Sequence-based reagent | PR-PCR 18 s Forward | Integrated DNA Technologies | 5'-GCTTAATTTGACTCAACACGGGA-3' | |
| Sequence-based reagent | PR-PCR 18 s Reverse | Integrated DNA Technologies | 5'-AGCTATCAATCTGTCAATCCTGTC-3' | |
| Sequence-based reagent | PR-PCR MTAP Forward-1 | Integrated DNA Technologies | 5'- ACGGCGGTGAAGATTGGAATA-3' | |
| Sequence-based reagent | PR-PCR MTAP Reverse-1 | Integrated DNA Technologies | 5'- ATGGCTTGCCGAATGGAGTAT –3' | |
| Sequence-based reagent | PR-PCR MTAP Forward-2 | Integrated DNA Technologies | 5'- AAGCCATCCGATGCCTTAATTT-3' | |
| Sequence-based reagent | PR-PCR MTAP Reverse-2 | Integrated DNA Technologies | 5'- TTGCCTGGTAGTTGACTTTTGAA –3' | |
| Commercial assay or kit | short tandem repeat (STR) profiling using PowerPlex 16 HS System | Promega | DC2320 | |
| Commercial assay or kit | Mycoplasma PCR Detection Kit | Sigma-Aldrich | MP0035 | |
| Commercial assay or kit | SuperSignal West Pico PLUS Chemiluminescent Substrate | Thermo Fisher Scientific | 34580 | |
| Commercial assay or kit | NEBNext HiFi 2 x PCR Master Mix | NEB | M0541L | |
| Commercial assay or kit | Mini-Elute PCR Purification Kit | QIAGEN | 28004 | |
| Commercial assay or kit | PowerUp SYBR Green master mix | Applied Biosystems | 4367659 | |
| Commercial assay or kit | SuperScript IV Reverse Transcriptase | Thermo Fisher Scientific | 18091050 | |

*Continued*

| Reagent type (species) or resource | Designation | Source or reference | Identifiers | Additional information |
|---|---|---|---|---|
| Commercial assay or kit | Kapa biosystems mouse genotyping extraction kit | Kapa biosystems | KK7352 | |
| Commercial assay or kit | TDE1,TAGMENT DNA ENZYME,24 RXN | Illumina | 15027865 | |
| Commercial assay or kit | RNeasy Plus Mini Kit (250) | QIAGEN | 74136 | |
| Commercial assay or kit | Seahorse XF Real-Time ATP Rate Assay Kit | Agilent | 103592–100 | |
| Commercial assay or kit | Seahorse XF Cell Mito Stress Test Kit | Agilent | 103015–100 | |
| Commercial assay or kit | NucleoBond Xtra EF kits | Takara Bio USA | 740424–50 | |
| Chemical compound, drug | Dulbecco's Modified Eagle Medium | Corning | 15–013-CV | |
| Chemical compound, drug | McCoy's 5 A Medium | Thermo Fischer Scientific | 16600082 | |
| Chemical compound, drug | L-Glutamine | Corning | A2916801 | |
| Chemical compound, drug | Fetal bovine serum | Sigma-Aldrich | A5256701 | |
| Chemical compound, drug | Penicillin/Streptomycin | Thermo Fischer Scientific | 15140122 | |
| Chemical compound, drug | XF calibrant | Agilent | 100840–000 | |
| Chemical compound, drug | XF DMEM medium | Agilent | 103575–100 | |
| Chemical compound, drug | Glucose | Agilent | 103577–100 | |
| Chemical compound, drug | pyruvate | Agilent | 103578–100 | |
| Chemical compound, drug | L-glutamine | Agilent | 103579–100 | |
| Chemical compound, drug | Tris HCl | BioWorld | 40120297–1 | |
| Chemical compound, drug | Dithiothreitol | Thermo Fischer Scientific | A39255 | |
| Chemical compound, drug | Bromophenol blue | Thermo Fischer Scientific | A18469.18 | |
| Chemical compound, drug | Sodium dodecyl sulfate | Thermo Fischer Scientific | BP166-500 | |
| Chemical compound, drug | Glycerol | Sigma | G5516-500ml | |
| Chemical compound, drug | TWEEN 20 | Bio-Rad | 1706531 | |
| Chemical compound, drug | Gibco PBS, pH 7.4 | Thermo Fischer Scientific | 10-010-072 | |
| Chemical compound, drug | Formaldehyde | Thermo Fischer Scientific | F79-1 | |

*Continued on next page*

*Continued*

| Reagent type (species) or resource | Designation | Source or reference | Identifiers | Additional information |
|---|---|---|---|---|
| Chemical compound, drug | Concanavalin A-coated beads | Bangs laboratories | BP531 | |
| Chemical compound, drug | KCl | Sigma-Aldrich | 58221–500 ml | |
| Chemical compound, drug | NP-40 | Thermo Fisher Scientific | 28324 | |
| Chemical compound, drug | Bovine Serum Albumin | Thermo Fisher Scientific | BP1605-100 | |
| Chemical compound, drug | Opti-MEM | ThermoFisher Scientific | 31985062 | |
| Chemical compound, drug | PEIpro transfection reagent | Polyplus | 101000033 | |
| Chemical compound, drug | HEPES | GIBCO | 15630–080 | |
| Chemical compound, drug | Proteinase K | Roche | 3115879001 | |
| Software, algorithm | Prism 9.0 | Prism 9.0 | Prism 9.0 | |
| Software, algorithm | Image J | Image J | https://imagej.nih.gov/ij/ | |
| Software, algorithm | Biorender | Biorender.com | https://www.biorender.com/ | |
| Software, algorithm | Adobe Illustrator 2024 | Adobe | https://www.adobe.com/it/products/illustrator.html | |
| Software, algorithm | SJCRH's Strongarm pipeline | Center for Applied Bioinformatics, St Jude Children's Research Hospital | N/A | |
| Software, algorithm | HOMER suite v4.8.3 | http://homer.ucsd.edu/homer/ | http://homer.ucsd.edu/homer/ | |
| Software, algorithm | Picard (version 1.65) | https://broadinstitute.github.io/picard/ | https://broadinstitute.github.io/picard/ | |
| Software, algorithm | MACS2 (version 2.2.7.1) | https://pypi.org/project/MACS2/ | https://pypi.org/project/MACS2/ | |
| Software, algorithm | SICER (version 1.1) | https://github.com/dariober/SICERpy (*Beraldi, 2017*) | https://github.com/dariober/SICERpy | |
| Software, algorithm | BEDtools version 2.25.0 | https://github.com/arq5x/bedtools2 (*Quinlan, 2025*) | https://github.com/arq5x/bedtools2 | |
| Software, algorithm | bedGraphToBigWig | https://hgdownload.soe.ucsc.edu/admin/exe/linux.x86_64.v369/ | https://hgdownload.soe.ucsc.edu/admin/exe/linux.x86_64.v369/ | |
| Software, algorithm | Integrated Genomics Viewer (IGV 2.3.82) | https://igv.org/doc/desktop/ | https://igv.org/doc/desktop/ | |
| Software, algorithm | EdgeR (version 3.16.5) | https://bioconductor.org/packages/edgeR | https://bioconductor.org/packages/edgeR | |
| Software, algorithm | GSEA | https://www.gsea-msigdb.org/gsea/index.jsp | https://www.gsea-msigdb.org/gsea/index.jsp | |
| Software, algorithm | Draw Venn Diagram | https://bioinformatics.psb.ugent.be/webtools/Venn/ | https://bioinformatics.psb.ugent.be/webtools/Venn/ | |

*Continued on next page*

*Continued*

| Reagent type (species) or resource | Designation | Source or reference | Identifiers | Additional information |
|---|---|---|---|---|
| Software, algorithm | Heatmap | https://www.bioinformatics.com.cn/en?keywords=heatmap | https://www.bioinformatics.com.cn/en?keywords=heatmap | |
| Software, algorithm | STRING | https://string-db.org/ | https://string-db.org/ | |
| Software, algorithm | Enrichr | https://maayanlab.cloud/Enrichr/ | https://maayanlab.cloud/Enrichr/ | |
| Software, algorithm | DepMAP | https://depmap.org/portal/ | https://depmap.org/portal/ | |
| Software, algorithm | cBioportal | https://www.cbioportal.org/ | https://www.cbioportal.org/ | |
| Software, algorithm | Seahorse software | https://www.agilent.com/en/product/cell-analysis/real-time-cell-metabolic-analysis/xf-software/agilent-seahorse-analytics-787485 | https://www.agilent.com/en/product/cell-analysis/real-time-cell-metabolic-analysis/xf-software/agilent-seahorse-analytics-787485 | |
| Software, algorithm | CRISPR algorithm | https://bitbucket.org/liulab/mageck-vispr/src/master/ | https://bitbucket.org/liulab/mageck-vispr/src/master/ | |
| Software, algorithm | Incucyte software | https://shop.sartorius.com/us/p/incucyte-spheroid-analysis-software-module/9600-0019 | https://shop.sartorius.com/us/p/incucyte-spheroid-analysis-software-module/9600-0019 | |
| Other (Deposited data) | RNA-seq | This paper | GSE240980 | |
| Other (Deposited data) | ATAC-seq | This paper | GSE262074 | |

## Established cell lines and spheroid culture

NEJF10, NEJF6 and HepG2 (ATCC, HB-8065) cells were maintained in DMEM (Thermo Fisher Scientific, Cat#MT10013CM) supplemented with 10% FBS (Gibco, Cat#10437028), and 1% Penicillin- Streptomycin solution (Gibco, Cat#15140122) at 37°C in 5% $CO_2$ in a humidified incubator. U2OS (ATCC, HTB96) and HCT116 (ATCC, CCL-247) were cultured with McCoy's 5 A (Thermo Fisher Scientific, 16600082) supplemented with 10% FBS (Gibco, Cat#10437028), and 1% Penicillin- Streptomycin solution (Gibco, Cat#15140122) at 37°C in 5% $CO_2$ in a humidified incubator. The spheroids formed when NEJF10 or NEJF6 liver cancer cell lines were cultured in low-attachment dish or flask with standard complete DMEM media as mentioned above for 2D cell culture.

## Modified cell lines

### Generation of NEJF10 cells with PRMT5 shRNA Knockdown

Plasmids of shRNA-ctrl (Vectorbuilder, VB010000-0009mxc), PRMT5-shRNA#1 (Vectorbuilder, VB900070-3948cva), PRMT5-shRNA#2 (Vectorbuilder, VB900070-3942srh), and PRMT5-shRNA#3 (Vectorbuilder, VB900080-7913zan) were purchased from Vector Builder. Plasmids were maxipreped by using NucleoBond Xtra EF kits (Takara Bio USA, 740424–50) following manufacturer's protocol. shRNA Lentivirus were produced with transient transfection of PEI-pro plasmids complex (6 µg of shRNA-ctrl, PRMT5shRNA#1, shRNA#2, shRNA#3, 3 µg of 1–1 r, 1 µg RTR, 1 µg of VSVg with 22 µl of PEI pro in 400 µl of DMEM medium) with 5x10⁶ HEK293T cells in 10 ml complete medium (DMEM, 10% FBS and 100 U/ml penicillin/streptomycin) in a 10 cm dish. Virus supernatant was collected every 8–12 hr for 3 days, which were passed through a 0.45 µm filter and concentrated by ultracentrifuge at 28,500 rpm for 2 hr at 4 °C. NEJF10 cell was seeded in six well plate 1 day before lentivirus transduction. The lentivirus particles were added to NEJF10 cells with polybrene to final concentration of 8 µg/ml. Puromycin (4 µg/ml in complete medium) selection were performed in the next day after virus transduction. After three days selection, NEJF10-shRNA-ctrl, NEJF10-PRMT5-shRNA#1, NEJF10-PRMT5-shRNA#2 and NEJF10-PRMT5-shRNA#3 post selection cells were cultured and expanded to

perform 2D crystal violet staining under hypoxia and normoxia culture conditions and 3D organoids proliferation assay with DMEM complete medium without puromycin.

## PRMT5 knockout in ABC-Myc genetic mouse model

*Albumin-Cre* (*Alb-Cre*) (Strain #003574), R26StopFLMyc(*CAG-Myc*) (Strain #020458), and *Prmt5* flox (Strain # 034414) mice were obtained from the Jackson Laboratory. To generate *Prmt5* specific KO in mouse liver, Alb-Cre$^{+/+}$ mice were bred with *Prmt5$^{flox/flox}$* mice to get *Alb-Cre$^{+/wt}$::Prmt5$^{flox/wt}$* mice. *CAG-Myc$^{+/+}$* mice were bred with *Prmt5$^{flox/flox}$* mice to obtain *CAG-Myc$^{+/wt}$::Prmt5$^{flox/wt}$* mice. Then, *Alb-Cre$^{+/wt}$::Prmt5$^{flox/wt}$* mice were bred with *CAG-Myc$^{+/wt}$::Prmt5$^{flox/wt}$* to generate *Alb-Cre$^{+/wt}$:: CAG-M$^{+/wt}$::Prmt5$^{wt/wt}$* (ABC-Myc), *Alb-Cre$^{+/wt}$:: CAG-Myc$^{+/wt}$::Prmt5$^{flox/wt}$* (*ABC-Myc-Prmt5$^{fl/wt}$*), *Alb-Cre$^{+/wt}$:: CAG-Myc$^{+/wt}$::Prmt5$^{fl/fl}$* (*ABC-MYC-Prmt5$^{fl/fl}$*), *Alb-Cre$^{+/wt}$::CAG-Myc$^{wt/wt}$::Prmt5$^{fl/wt}$* (*Alb-Cre-Prmt5$^{fl/wt}$*), *Alb-Cre$^{+/wt}$::CAG-Myc$^{wt/wt}$::Prmt5$^{fl/fl}$* (*Alb-Cre-Prmt5$^{fl/fl}$*). For genotyping, the genomic DNA was extracted from tail biopsies, and PCR amplification assay was performed using KAPA Mouse Genotyping Kits (Roche Corporate, Cat#KK7352) according to The Jackson Laboratory genotyping PCR conditions for each mice strain. The primers 5'-TGC AAA CAT CAC ATG CAC AC, GAA GCA GAA GCT TAG GAA GAT GG-3' and 5'-TTG GCC CCT TAC CAT AAC TG-3' were used for Alb-Cre genotyping (AlbCre = 390 bp and WT = 351 bp). The primers 5'-CCA AAG TCG CTC TGA GTT GTT ATC-3', 5'-GAG CGG GAG AAA TGG ATA TG-3', 5'-CCA AGA GGG TCA AGT TGG A-3' and 5'-GCA ATA TGG TGG AAA ATA AC-3' are used for CAG-Myc genotyping (MYC = 550 bp and WT = 604 bp). The primers 5'- GAT ACC AGG ACC CAC ACT GG –3', and 5'- CTT AGG GGC TTC ATG CAT CT-3' were used for Prmt5 flox genotyping (flox ~330 bp and wild type = 239 bp). The genotyping PCR products were resolved in 2% agarose gel (Invitrogen, Cat#16500–500) and imaged with Li-COR D-Digit (Li-COR, 3500). Mice were housed with temperature and 12 hr light /12 hr dark cycle controlled under specific-pathogen-free conditions (SPF) at the St Jude Children's Research Hospital mouse facility. All experiments that involved the use of mice were performed in accordance with the guidelines outlined by the St Jude Children's Research Hospital Institutional Animal Care and Use Committee (IACUC; approved protocol #615).

## Method details

### Seahorse real-time ATP rate assay

The ATP production rate assay was determined using the Seahorse XF Real-Time ATP Rate Assay Kit (Agilent, 103592–100) and the Seahorse XF Pro Analyzer (Agilent). Briefly, cells were seeded into the Seahorse XF Pro M cell culture microplate (Agilent, 103774–100) at different cell densities (10000, 20000, and 4000 cells per well). At the same time, the XF Pro Sensor Cartridge was hydrated using 200 µl of XF calibrant (Agilent, 100840–000) in a 37 °C CO2-free incubator overnight. The next day, the cells were washed and incubated with XF DMEM medium (Agilent,103575–100) supplemented with 10 mM glucose (Agilent, 103577–100), 1 mM pyruvate (Agilent, 103578–100), and 2 mM L-glutamine (Agilent, 103579–100). OCR and ECAR were measured in the Agilent's Seahorse XF Pro Extracellular Flux Analyzer by subsequent sequential injections of two compounds that affect the cellular bioenergetic processes, as follows: 20 µl of oligomycin (10 µM) in port A and 22 µl of rotenone/antimycin A (5 µM) in port B. Data was processed with Seahorse Analytics.

### RNA-seq analysis for differential gene expression (DGE)

NEJF10 cells were cultured with DMEM complete medium in 6 well plate (Falcon,353046) for 2D and in ultralow attachment 6 well plate (Corning, 3471) for 3D under 21% and 1% oxygen conditions for 48 hr. Total RNA extraction from cells was performed using the RNeasy Mini Kit (74106, Qiagen) according to the manufacturer's instructions.

Total stranded RNA sequencing data were processed by the internal AutoMapper pipeline. Briefly the raw reads were first trimmed (Trim-Galore version 0.60), mapped to mouse genome assembly (mm10; STAR v2.7) and then the gene level values were quantified (RSEM v1.31) based on GENCODE annotation (M22). Low count genes were removed from analysis using a CPM cutoff corresponding to a count of 10 reads and only confidently annotated (level 1 and 2 gene annotation) and protein-coding genes are used for differential expression analysis. Normalization factors were generated using the TMM method, counts were normalized using voom and normalized counts were analyzed using the lmFit and eBayes functions (R limma package version 3.6.3). The significantly up- and

down- regulated genes were defined by at least twofold changes and adjusted p-value <0.05. Then Gene Set Enrichment Analysis (GSEA) was conducted using gene-level log2 fold changes from differential expression results against gene sets in the Molecular Signatures Database (MSigDB 6.2; gsea2 version 2.2.3).

## RNA-seq analysis for alternative splicing analysis

After mapping RNA-seq data, rMATS v4.1.0 was used for RNA alternative splicing analysis by using the mapped BAM files as input. Specifically, five different kinds of alternative splicing events were identified, that is skipped exon (SE), alternative 5'-splicing site (A5SS), alternative 3'-splicing site (A3SS), mutually exclusive exon (MXE) and intron retention (RI). To keep consistent, the same GTF annotation reference file for mapping was used for rMATS. For stranded RNA-seq data, the argument '--libType fr-firststrand' was applied. To process reads with variable lengths, the argument '--variable-read-length' was also used for rMATS. To select statistically significantly differential splicing events, the following thresholds were used: FDR <0.05 and the absolute value of IncLevel-Difference >0.1. For visualization, the IGV Genome Browser was used to show the sashimi plots of splicing events.

## CRISPR screening and data analysis

The Mouse CRISPR Knockout Pooled Library (Brie, lentiCRISPRv2) was obtained from Addgene (Addgene#73632), which includes 1000 control gRNAs and 78,637 unique sgRNA sequences targeting 19,674 human genes (4 sgRNAs per gene, and 1000 non-targeting controls). The plasmid library was amplified and validated in the Center for Advanced Genome Engineering at St. Jude Children's Research Hospital as described in the Broad GPP protocol (here) except EnduraTM DUOs (Lucigen) electrocompetent cells were used for the transformation step. The workflow of this whole genome genetic screen is illustrated in *Figure 2A*. The NEJF10 cells were transduced with mouse CRISPR Knockout pooled library (Brie) at a low MOI (~0.3) to ensure effective barcoding of individual cells. Cells were replenished with fresh DMEM medium containing 2 µg/ml puromycin (Millipore Sigma) for 36 hr. After puromycin selection, cells were washed to eliminate dead cell debris and maintained in complete DMEM medium.

In the process of 2D screening, to maintain adequate representation of gRNAs, during the CRISPR screening, a total of 8 million cells were re-plated during each time point, ensuring a consistent 100 x coverage of the Brie library. The 2D hypoxia screening took place within a hypoxia chamber, and all cell culture activities were conducted under the controlled environment of this chamber (Whitley H35 HEPA Hypoxystation). In the context of Brie-library screening involving NEJF-10 cells within a 3D culture framework, 0.5 million cells were introduced into each well of a 96-well plate (non-adherent), totaling 144 wells. A centrifugation step at 1000 rpm for 5 min was subsequently undertaken. Each individual spheroid maintained an approximate coverage of ~6.2 x for each gRNA, and when combined, the total gRNA coverage across all spheroids reached 900 x. On the following day, the formed spheroids were carefully transferred to a non-adherent T75 flask, ensuring their spherical structure was preserved. The spheroids were cultured within a regular $CO_2$ incubator, utilizing a shaker operating at 80 rpm. To maintain their growth, the cell culture medium was replenished every alternate day. At each time point $32 \times 10^6$ cells were collected for genomic DNA extraction to ensure over 400×coverage.

The total genomic DNA was extracted using a DNeasy Blood & Tissue Kit (QIAGEN) and quantified with a Nanodrop instrument. The sgRNA sequences were amplified using PCR method using NEB Q5 polymerase (New England Biolabs). PCR products were purified by AMPure XP SPRI beads (Beckman Coulter) and quantified by a Qubit dsDNA HS assay (Thermo Fisher Scientific). A total of 16 million reads were sequenced using an Illumina HiSeq sequencer, and the sequencing data were analyzed using MAGeCK-VISPR software (*Li et al., 2015*). NGS sequencing was performed in the Hartwell Center Genome Sequencing Facility at St. Jude Children's Research Hospital. Single-end, 100-cycle sequencing was performed on a NovaSeq 6000 (Illumina). Validation to check gRNA presence and representation was performed using calc_auc_v1.1.py (https://github.com/mhegde/auc-calculation; *Hedge, 2020*) and count_spacers.py. Network analysis performed using STRING program (https://string-db.org/).

## Lentivirus preparation and transduction for VHL knockout in HepG2 cells

The lentiviral-based knockout plasmids targeting the VHL gene (VHL-CRISPR guide RNA1 pLentiCRISPR v2 and VHL-CRISPR guide RNA2 pLentiCRISPR v2, Cat#SC1678) were procured from GenScript. The control plasmid, pLentiCRISPR v2 (Cat#52961), was sourced from Addgene. Plasmids were maxiprepped by using NucleoBond Xtra EF kits (Takara Bio USA, 740424–50) following manufacturer's protocol. Lentivirus was produced by transient transfection of PEI-pro DNA complex (LentiV2, VHL-gRNA-1 and VHL-gRNA-2 plasmids, 1–1 r, RTR, VSVg with PEI pro in DMEM medium) with HEK293T cells in 12 ml DMEM complete medium in a 15 cm dish. Virus supernatant was collected every 8–12 hr for 3 days, which were passed through a 0.45 µm filter and concentrated by ultracentrifuge at 28,500 rpm for 2 hr at 4 °C. The VHL-gRNA virus particles were added to HepG2 cells which seeded previous day, following add polybrene to final concentration of 8 µg/ml. Puromycin (1 µg/ml in complete medium) selection were performed in the next day after virus transduction. At the end of the 72 hr, all HepG2 parental control cells were dead. HepG2-lentiV2, HepG2-VHL-KO1 and HepG2-VHL-KO2 cells post selection were cultured and expanded with DMEM complete medium. For 2D and 3D cell proliferation assay, HepG2-lentiV2, HepG2-VHL-KO1 and HepG2-VHL-KO2 cells were cultured with 96-well clear flat bottom (Corning, 353072) for 2D and 96-well clear round bottom ultra-low attachment plate (Corning, 7007) for 3D in the IncuCyte. High resolution bright field images were captured every 6–8 hr. Analysis modules use edge finding algorithm to quantity 2D and 3D growth and size over time to measure cell proliferation. The raw data of 2D cell proliferation were exported to and plotted with GraphPad Prism 9.5.1. For the 3D cell proliferation, the raw data was exported to excel and normalized to 24 hr 3D size, then plotted graph with GraphPad Prism 9.5.1.

## Online bioinformatics tools and programs

Pathway analysis was performed by using Enrichr program (https://maayanlab.cloud/Enrichr/). Protein-protein interaction network was analyzed by using STRING program (https://string-db.org/). Correlation of knockout effects of two genes, knockout of one gene vs gene expression, gene knockout effect vs drug effect, gene expression vs drug effect was analyzed by DepMap data (https://depmap.org/portal/), then data were downloaded and presented by using PRISM program.

## Histology and immunohistochemistry

Liver tumors were fixed in 10% neutral buffered formalin, embedded in paraffin, sectioned at 4 µm, mounted on positive charged glass slides (Superfrost Plus; 12-550-15, Thermo Fisher Scientific, Waltham, MA) that were dried at 60 °C for 20 min, and stained with hematoxylin and eosin (HE). The following immunohistochemistry protocol was used for the detection of anti-Galectin-3 (Mac-2; M3/38, ACL8942AP, Accurate Chemical and Scientific Corporation), 1:1000, 32' incubation, heat-induced epitope retrieval with cell conditioning media 1 (Ventana Medical Systems, Tucson, AZ), 32 min; Visualization with DISCOVERY OmniMap anti-Rt HRP (760–4311; Ventana Medical Systems), DISCOVERY ChromoMap DAB kit (760-159; Ventana Medical Systems).

## Western blot

For western blotting, samples were mixed with equal volume of 2 X sample buffer (1 M TRIS/HCl, 10% SDS, 0.1% bromophenol-blue, 10% β−mercaptoethanol, 10% glycerol), sonicated for 30 s on 30% AMPL (Sonics & Materials Inc, VCX 130PB) and heated for 15 min at 95 °C. Proteins were resolved on SDS-polyacrylamide gels (Bio-Rad, Cat#4568083) and transferred onto PVDF membrane (Bio-Rad, Cat#170–4272) with Trans-blot Turbo transfer system (Bio-Rad, Cat#1704150). After being incubated with primary antibodies overnight at –20, horseradish peroxidase-(HRP) conjugated secondary antibody (Thermo Fisher, A245373, A245182) at 1: 5000 was used for 1 hr incubation. The signals were detected by chemiluminescence (ECL, Thermo Fisher). Images were taken using Li-COR Odyssey FC (Li-COR, Cat#2800). Antibodies including β-Actin (Sigma, A1978, 1:5,000), HIF1α (CAYMAN, 10006421, 1:200), HIF2α (Novus, NB100-122, 1:500), VHL (Cell Signaling Technology, 68547 S, 1:1000), PRMT5 (ABclonal, A1520, 1:1000), HSP90 (Santa Cruz, sc13119, 1:1000) and MTAP (Cell Signaling Technology, 4158 S,1:1000) were used for western blot.

## Crystal violet staining

NEJF10-shRNA-ctrl, NEJF10-PRMT5-shRNA#1, NEJF10-PRMT5-shRNA#2 and NEJF10-PRMT5-shRNA#3 was seeded in six-well plate (Falcon,353046) and cultured for 6 days. After removing

medium, cells were washed with Dulbecco's phosphate buffered saline without calcium or magnesium (DPBS, Lonza) and treated with 4% Formaldehyde in PBS (PFA) for 20 min. Once PFA was removed, cells were stained with 0.1% crystal violet stain for 1 hr, followed by water rinsing three times. The crystal violet staining plates were imaged after the plates were completely dry.

## Organoids proliferation assay of PRMT5 knockdown in NEJF10 cell organoids

NEJF10-shRNA-ctrl, NEJF10-PRMT5-shRNA#1, NEJF10-PRMT5-shRNA#2 and NEJF10-PRMT5-shRNA#3 3000 cells per well were cultured with 96-well clear round bottom ultra-low attachment plate (Corning, 7007) in the IncuCyte. IncuCyte scanning was used to live imaging and measure the proliferation and dynamics of PRMT5 knockdown in NEJF10 cell organoids. Replicates were 6–8 per group. High resolution bright field images were captured every 6–8 hr without human interaction. The organoids proliferation was analyzed and quantified using IncuCyte software. Analysis modules use edge finding algorithm to quantity organoid growth and size over time to measure organoid proliferation. Real-time analysis is performed over 9 days. The raw data were exported and plotted with GraphPad Prism 9.5.1.

## RT-PCR for MTAP

Total RNA from NEJF10 cells cultured under 2D normoxia, hypoxia and 3D normoxia conditions were performed using the RNeasy Mini Kit (QIAGEN) according to the manufacturer's instructions. cDNA was prepared from extracted RNA using Invitrogen SuperScript IV Reverse Transcriptase (Thermo Fisher Scientific, 18091050) and detected by fast SYBR Green (Applied Biosystems, 4368708) on Applied Biosystems 7500 Fast Real-time PCR System. Two sets of MTAP primers (set1: Forward 5'-ACGGCGGTGAAGATTGGAATA –3' and Reverse 5'- ATGGCTTGCCGAATGGAGTAT –3' and Set2: Forward 5'- AAGCCATCCGATGCCTTAATTT –3' and Reverse 5'- TTGCCTGGTAGTTGACTTTTGAA –3') were used for RT-PCR and 18 s primers (Forward 5'-GCTTAATTTGACTCAACACGGGA-3' and Reverse 5'-AGCTATCAATCTGTCAATCCTGTC –3') were performed as internal controls. qPCR signal for each gene was normalized to those of 18 s using the ΔCT method. Results were represented as fold expression relative to WT with the standard error for three to four biological replicates.

## ATAC-seq

NEJF10 cells were cultured in 21% and 1% oxygen in monolayer, or 21% oxygen in 3D spheroid. Fresh cultured NEJF10 cells (100,000 per sample) under different culture conditions were harvested and washed with 150 µl cold Dulbecco's Phosphate-Buffered Saline (DPBS) containing protease inhibitor (PI). Nuclei were collected by centrifuging at 500 × $g$ for 10 min at 4 °C after cell pellets were resuspended in lysis buffer (10 mM Tris-Cl pH 7.4, 10 mM NaCl, and 3 mM MgCl$_2$ containing 0.1% NP-40 and PI). Nuclei were incubated with Tn5 transposon enzyme in transposase reaction mix buffer (Illumina) for 30 min at 37 °C. DNAs were purified from transposition sample by using Min-Elute PCR purification kit (QIAGEN, Valencia, CA) and measured by Qubit. Polymerase chain reaction (PCR) was performed to amplify with High-Fidelity 2 X PCR Master Mix [72 °C /5min +98 °C /30 s +12 × (98 °C /10 s + 63 °C /30 s + 72 °C /60 s) + 72 °C /5 min]. The libraries were purified using Min-Elute PCR purification kit (QIAGEN, Valencia, CA). ATAC-seq libraries followed by pair-end sequencing on HiSeq4000 (Illumina) in the Hartwell Center at St Jude Children's Research Hospital. The ATAC-seq raw reads were aligned to the mouse reference genome (mm10) using BWA **Li and Durbin, 2010** to and then marked duplicated reads with Picard (version 1.65), with only high-quality reads kept by samtools (version 1.3.1, parameter '-q 1 F 1024'; **Li et al., 2009**). Reads mapping to mitochondrial DNA were excluded from the analysis. All mapped reads were offset by +4 bp for the +strand and –5 bp for the – strand (**Buenrostro et al., 2013**). Peaks were called for each sample using MACS2 (**Zhang et al., 2008**) with parameters '-q 0.01 –nomodel –extsize 200 –shift 100'. Peaks were merged for the same cell types using BEDtools (**Quinlan and Hall, 2010**). Peak annotation was performed using HOMER (**Heinz et al., 2010**). All sequencing tracks were viewed using the Integrated Genomic Viewer (IGV 2.3.82; **Thorvaldsdóttir et al., 2013**).

## Statistical analysis

All quantitative data are presented as mean ± SD. Unpaired Student's t test was performed for comparison of two groups. Kaplan-Meier method was used to estimate the survival rate. p-values across multiple time points were adjusted for multiple comparison using the Holm-SidaK method. $p < 0.05$ was considered as statistically significant. All the statistical analyses, except where otherwise noted, were performed using GraphPad Prism (v9).

## Acknowledgements

We thank the staff of the St. Jude Center for In Vivo Imaging and Therapeutics and Hartwell Center for their dedication and expertise. We thank Dr. Dongli Hu for mycoplasma testing and STR assay. This work was partly supported by American Cancer Society-Research Scholar (130421-RSG-17-071-01-TBG, JY) and National Cancer Institute (1R03CA212802-01A1, 1R01CA229739-01, 1R01CA266600-01A1, 1R0CA289881. JY). This work was supported by the American Lebanese Syrian Associated Charities (ALSAC). The content is solely the responsibility of the authors and does not necessarily represent the official views of the National Institutes of Health. The authors have declared that no conflict of interest exists.

## Additional information

### Funding

| Funder | Grant reference number | Author |
|---|---|---|
| American Cancer Society | 130421-RSG-17-071-01-TBG | Jun Yang |
| National Cancer Institute | 1R03CA212802 | Jun Yang |
| National Cancer Institute | 1R01CA229739 | Jun Yang |
| National Cancer Institute | 1R01CA266600 | Jun Yang |
| National Cancer Institute | 1R01CA289881 | Jun Yang |

The funders had no role in study design, data collection and interpretation, or the decision to submit the work for publication.

### Author contributions

Jie Fang, Shivendra Singh, Brennan Wells, Data curation, Formal analysis, Validation, Investigation, Visualization, Methodology; Qiong Wu, Data curation, Validation, Methodology; Hongjian Jin, Data curation, Formal analysis, Visualization, Methodology; Laura J Janke, Data curation, Formal analysis, Investigation, Visualization, Methodology; Shibiao Wan, Formal analysis; Jacob A Steele, Jon P Connelly, Data curation, Formal analysis, Methodology; Andrew J Murphy, Ruoning Wang, Andrew Davidoff, Margaret Ashcroft, Conceptualization, Writing – review and editing; Shondra M Pruett-Miller, Formal analysis, Supervision, Methodology, Writing – review and editing; Jun Yang, Conceptualization, Data curation, Formal analysis, Supervision, Funding acquisition, Investigation, Visualization, Methodology, Writing – original draft, Project administration

### Author ORCIDs

Jie Fang ⓘ http://orcid.org/0000-0001-6480-134X
Qiong Wu ⓘ https://orcid.org/0000-0002-6063-0800
Jacob A Steele ⓘ https://orcid.org/0000-0001-9924-2226
Andrew J Murphy ⓘ https://orcid.org/0000-0001-6747-0355
Ruoning Wang ⓘ https://orcid.org/0000-0001-9798-8032
Margaret Ashcroft ⓘ https://orcid.org/0000-0002-0066-3707
Shondra M Pruett-Miller ⓘ https://orcid.org/0000-0002-3793-585X
Jun Yang ⓘ https://orcid.org/0000-0002-4233-3220

### Ethics

This study was performed in strict accordance with the recommendations in the Guide for the Care and Use of Laboratory Animals of the National Institutes of Health. All of the animals were handled according to approved institutional animal care and use committee (IACUC) protocols (#615) of St Jude Children's Research Hospital.

Reviewer #1 (Public review): https://doi.org/10.7554/eLife.101299.3.sa1
Author response https://doi.org/10.7554/eLife.101299.3.sa2

## Additional files

### Supplementary files

Supplementary file 1. Differentially expressed genes for NEJF10 in hypoxia vs normoxia 2D culture.

Supplementary file 2. Differentially expressed genes for NEJF10 in 3D vs 2D culture in normoxia.

Supplementary file 3. Analysis of five major splicing events (2D Normoxia vs 2D Hypoxia).

Supplementary file 4. Analysis of five major splicing events (2D Normoxia vs 3D Normoxia).

Supplementary file 5. Analysis of five major splicing events (3D Normoxia vs 2D Hypoxia).

Supplementary file 6. KEGG pathway analysis for the spliced genes (2D Hypoxia vs 2D Normoxia).

Supplementary file 7. KEGG pathway analysis for the spliced genes (3D Normoxia vs 2D Normoxia).

Supplementary file 8. KEGG pathway analysis for the spliced genes (3D Normoxia vs 2D Hypoxia).

Supplementary file 9. NEJF10 CRISPR results with Venn analysis.

Supplementary file 10. NEJF6 CRISPR results.

Supplementary file 11. Venn analysis for the differentially expressed genes and fitness genes cultured under different conditions.

MDAR checklist

### Data availability

Sequencing data have been deposited in GEO under accession codes GSE240980 and GSE262074. All data generated or analysed during this study are included in the manuscript and supporting files including *Supplementary files 1–11*.

The following datasets were generated:

| Author(s) | Year | Dataset title | Dataset URL | Database and Identifier |
|---|---|---|---|---|
| Fang J, Wan S, Jin H, Yang J | 2024 | Cellular fitness of MYC-driven cancer cells to genetic and pharmacologic perturbations in normoxia, hypoxia and 3D | https://www.ncbi.nlm.nih.gov/geo/query/acc.cgi?&acc=GSE240980 | NCBI Gene Expression Omnibus, GSE240980 |
| Fang J, Wan S, Jin H, Yang J | 2024 | Cellular fitness of MYC-driven cancer cells to genetic and pharmacologic perturbations in normoxia, hypoxia and 3D | https://www.ncbi.nlm.nih.gov/geo/query/acc.cgi?&acc=GSE262074 | NCBI Gene Expression Omnibus, GSE262074 |

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
