## [Editor Report · eLife Assessment]

This manuscript provides potentially **important** findings examining in 2D and 3D models in MYC liver cancer cells changes in DNA repair genes and programs in response to hypoxia. The authors use **convincing** methodology in most cases, but there is some concern that the analysis is **incomplete**.

---

## [Referee Report · Reviewer #1 (Public review)]

Summary:

In this report, the authors made use of a murine cell line derived from a MYC-driven liver cancer to investigate the gene expression changes that accompany the switch from normoxic to hypoxia conditions during 2D growth and the switch from 2D monolayer to 3D organoid growth under normoxic conditions. They find a significant (ca. 40-50%) overlap among the genes that are dysregulated in response to hypoxia in 2D cultures and in response to spheroid formation. Unsurprisingly, hypoxia-related genes were among the most prominently deregulated under both sets of conditions. Many other pathways pertaining to metabolism, splicing, mitochondrial electron transport chain structure and function, DNA damage recognition/repair and lipid biosynthesis were also identified.

Comments on the revised manuscript:

In my original review of this manuscript, I raised 11 points that I thought needed to be addressed and/or clarified by the authors. In response, they have provided an adequate answer to only one of these (point 6), which is little more than a more thorough description of how spheroids were generated. The remaining points that I raised, which would have provided more mechanistic insight into their study were addressed by the authors with the following such comments:

- It is not the focus of this study (Points 1 and 4)

- It is worthy of further validation (Point 2)

- We apologize for not being able to validate everything (Point 3)

- This reviewer has raised an interesting question. We are investigating this hypothesis and hopefully we can give a clear answer in the future (Point 5)

- This is an excellent idea that we certainly will do it in our future experiments (Point 7)

As to responses that the authors made to the other two reviewers' comments: Most pertained to cosmetic alterations involving clarification of methods, inclusion of a new figure or rearrangement of old figures. These were generally answered. However, in response to the last point raised by Rev. 3 to compare "sgRNA abundances at the earliest harvesting time with the distribution in the library...to see whether and to what extent selection has already taken place before the three culture conditions were established", the authors responded with the comment: "This is great point. Unfortunately, we did not perform such an analysis."

I understand that it is often impossible to address all points raised by the reviewers. This can be for a variety of reasons and many times the omissions can be overlooked and accepted if the reviewer can be convinced that a good faith attempt has otherwise been made to address the other deficiencies. However, no such effort has been made here and the study remains deficient and largely descriptive as I pointed out in my original review.

---

## [Author Response]

The following is the authors’ response to the original reviews

**Public Reviews:**

**Reviewer #1 (Public review):**
Summary:In this report, the authors made use of a murine cell life derived from a MYC-driven liver cancer to investigate the gene expression changes that accompany the switch from normoxic to hypoxia conditions during 2D growth and the switch from 2D monolayer to 3D organoid growth under normoxic conditions. They find a significant (ca. 40-50%) overlap among the genes that are dysregulated in response to hypoxia in 2D cultures and in response to spheroid formation. Unsurprisingly, hypoxia-related genes were among the most prominently deregulated under both sets of conditions. Many other pathways pertaining to metabolism, splicing, mitochondrial electron transport chain structure and function, DNA damage recognition/repair, and lipid biosynthesis were also identified.

We thank this reviewer for his/her time and efforts, and the insightful comments.

Major comments:(1) Lines 239-240: The authors state that genes involved in DNA repair were identified as being necessary to maintain survival of both 2D and 3D cultures (Figure S6A). Hypoxia is a strong inducer of ROS. Thus, the ROS-specific DNA damage/recognition/repair pathways might be particularly important. The authors should look more carefully at the various subgroups of the many genes that are involved in DNA repair. They should also obtain at least a qualitative assessment of ROS and ROS-mediated DNA damage by staining for total and mitochondrial-specific ROS using dyes such as CM-H2-DCFDA and MitoSox. Actual direct oxidative damage could be assessed by immunostaining for 8-oxo-dG and related to the sub-types of DNA damage-repair genes that are induced. The centrality of DNA damage genes also raises the question as to whether the previously noted prominence of the TP53 pathway (see point 5 below) might represent a response to ROS-induced DNA damage.

We thank this reviewer for the insightful comments, and agreed that ROS induced by hypoxia could play a role in modulating DNA repair and consequently cellular essentiality. Although pathway enrichment in Figure S6A (now as Figure 2-figure supplement 4A) showed that DNA repair pathway was essential to cell survival in hypoxia and 3D cultures, the genes associated with this pathway (Ddb1;Brf2;Gtf3c5;Guk1;Taf6) are not typical DNA repair genes. They are more likely involved in gene transcription. However, it will be interesting to see if they are specifically involved in DNA damage in response to ROS, which is out of focus of this study.

(2) Because most of the pathway differences that distinguish the various cell states from one another are described only in terms of their transcriptome variations, it is not always possible to understand what the functional consequences of these changes actually are. For example, the authors report that hypoxia alters the expression of genes involved in PDH regulation but this is quite vague and not backed up with any functional or empirical analyses. PDH activity is complex and regulated primarily via phosphorylation/dephosphorylation (usually mediated by PDK1 and PDP2, respectively), which in turn are regulated by prevailing levels of ATP and ADP. Functionally, one might expect that hypoxia would lead to the down-regulation of PDH activity (i.e. increased PDH-pSer392) as respiration changes from oxidative to non-oxidative. This would not be appreciated simply by looking at PDH transcript levels. This notion could be tested by looking at total and phospho-PDH by western blotting and/or by measuring actual PDH activity as it converts pyruvate to AcCoA.

We agreed with this reviewer that PDH activity regulation could be affected by multi-factors, and it is worthy of further validation by other approaches.

(3) Line 439: Related to the above point: the authors state: "It is likely that blockade of acetyl-CoA production by PDH knockout may force cells to use alternative energy sources under hypoxic and 3D conditions, averting the Warburg effect and promoting cell survival under limited oxygen and nutrient availability in 3D spheroids." This could easily be tested by determining whether exogenous fatty acids are more readily oxidized by hypoxic 2D cultures or spheroids than occurs in normoxic 2D cultures.

We thank for this suggestion. We apologized for not being able to validate everything.

(4) Line 472: "Hypoxia induces high expression of Acaca and Fasn in NEJF10 cells indicating that hypoxia promotes saturated fatty acid synthesis...The beneficial effect of Fasn and Acaca KO to NEJF10 under hypoxia is probably due to reduction of saturated fatty acid synthesis, and this hypothesis needs to be tested in the future.". As with the preceding comment, this supposition could readily be supported directly by, for example, performing westerns blots for these enzymes and by showing that incubation of hypoxic 2D cells or spheroids converted more AcCoA into lipid.

We thank for this suggestion. However, functional validation for the Fasn and Acaca KO is out of focus in this study.

(5) In Supplementary Figure 2B&C, the central hub of the 2D normoxic cultures is Myc (as it should well be) whereas, in the normoxic 3D, the central hub is TP53 and Myc is not even present. The authors should comment on this. One would assume that Myc levels should still be quite high given that Myc is driven by an exogenous promoter. Does the centrality of TP53 indicate that the cells within the spheroids are growtharrested, being subjected to DNA damage and/or undergoing apoptosis?

The predicted transcription factor activity analysis was based on the differential ATAC-seq peaks among different culture through pairwise comparisons. If TP53 and MYC were not present under that condition, it did not mean their activity was absent.

“…the centrality of TP53 indicate that the cells within the spheroids are growth-arrested, being subjected to DNA damage and/or undergoing apoptosis?” This reviewer has raised an interesting question. We are investigating this hypothesis and hopefully we can give a clear answer in the future.

(6) In the Materials and Methods section (lines 711-720), the description of how spheroid formation was achieved is unclear. Why were the cells first plated into non-adherent 96 well plates and then into nonadherent T75 flasks? Did the authors actually utilize and expand the cells from 144 T75 flasks and did the cells continue to proliferate after forming spheroids? Many cancer cell types will initially form monolayers when plated onto non-adherent surfaces such as plastic Petri dishes and will form spheroid-like structures only after several days. Other cells will only aggregate on the "non-adherent" surface and form spheroid-like structures but will not actually detach from the plate's surface. Have the authors actually documented the formation of true, non-adherent spheroids at 2 days and did they retain uniform size and shape throughout the collection period? The single photo in Supplementary Figure 1 does not explain when this was taken. The authors include a schematic in Figure 2A of the various conditions that were studied. A similar cartoon should be included to better explain precisely how the spheroids were generated and clarify the rationale for 96 well plating. Overall, a clearer and more concise description of how spheroids were actually generated and their appearance at different stages of formation needs to be provided.

The cells were initially plated in non-adherent 96-well plates to facilitate the formation of spheroids in a controlled and uniform manner. As correctly mentioned by the reviewer, during the initial stages, cells cultured on non-adherent surfaces often form aggregates or clumps, and it takes a few days for them to develop into solid spheroids.

In our study, we aimed to achieve 3D spheroid formation immediately following the transduction process to allow for screening under both 2D and 3D conditions. Plating the cells into 96-well plates enabled us to monitor and control the formation of spheroids in smaller volumes before scaling up the culture in non-adherent T75 flasks for subsequent experimental steps. This setup allows us to maintain gene editing processes under both 2D and 3D conditions.

Regarding the proliferation and uniformity of spheroids:

• Yes, the spheroids continued to proliferate after their formation.

• True, non-adherent spheroids were documented as early as the next day. This was visually confirmed under microscopy, and size uniformity was maintained throughout the collection period by following optimized culture protocols.

We also agreed with the reviewer’s suggestion to include a cartoon schematic similar to Figure 2A, illustrating the spheroid generation process and clarifying the rationale for using 96-well plates. We have included such a cartoon and speroid growth curve monitored by Incucyte as Figure 2-figure supplement 2.

(7) The authors maintained 2D cultures in either normoxic or hypoxic (1% O2) states during the course of their experiments. On the other hand, 3D cultures were maintained under normoxic conditions, with the assumption that the interiors of the spheroids resemble the hypoxic interiors of tumors. However, the actual documentation of intra-spheroid hypoxia is never presented. It would be a good idea for the authors to compare the degree of hypoxia achieved by 2D (1% O2) and 3D cultures by staining with a hypoxia-detecting dye such as Image-iT Green. Comparing the fluorescence intensities in 2D cultures at various O2 concentrations might even allow for the construction of a "standard curve" that could serve to approximate the actual internal O2 concentration of spheroids. This would allow the authors to correlate the relative levels of hypoxia between 2D and 3D cultures.

This is an excellent idea that we certainly will do it in our future experiments.

(8) Related to the previous 2 points, the authors performed RNAseq on spheroids only 48 hours after initiating 3D growth. I am concerned that this might not have been a sufficiently long enough time for the cells to respond fully to their hypoxic state, especially given my concerns in Point 6. Might the results have been even more robust had the authors waited longer to perform RNA seq? Why was this short time used?

We agreed with this reviewer. We were unsure if 48hours was an ideal timepoint. It might be necessary to perform a longitudinal experiment to harvest samples under different timepoints in the future experiments.

(9) What happens to the gene expression pattern if spheroids are re-plated into standard tissue culture plates after having been maintained as spheroids? Do they resume 2D growth and does the gene expression pattern change back?

This is a great question and we have never thought about what the gene expression pattern would be if speroids are re-plated in 2D. This could be a challenging experiment because the gene expression and epigenetic changes are timing related. However, the cells do grow well after re-plated in 2D.

(10) Overall, the paper is quite descriptive in that it lists many gene sets that are altered in response to hypoxia and the formation of spheroids without really delving into the actual functional implications and/or prioritizing the sets. Some of these genes are shown by CRISPR screening to be essential for maintaining viability although in very few cases are these findings ever translated into functional studies (for example, see points 14 above). The list of genes and gene pathways could benefit from a better explanation and prioritization of which gene sets the authors believe to be most important for survival in response to hypoxia and for spheroid formation.

This was a genome-wide study that integrated RNA-seq, ATAC-seq and CRISPR KO, providing resource to understand the oncogenic pathways in different culture conditions. We believe we have clearly articulated the important genes/pathways in our abstract.

(11) The authors used a single MYC-driven tumor cell line for their studies. However, in their original paper (Fang, et al. Nat Commun 2023, 14: 4003.) numerous independent cell lines were described. It would help to know whether RNAseq studies performed on several other similar cell lines gave similar results in terms of up & down-regulated transcripts (i.e. representative of the other cell lines are NEJF10 cells).

We have not generated RNA-seq data for these cell lines cultured in different conditions.

**Reviewer #2 (Public review):**
Summary:The manuscript by Fang et al., provides a tour-de-force study uncovering cancer cell's varied dependencies on several gene programs for their survival under different biological contexts. The authors addressed genomic differences in 2D vs 3D cultures and how hypoxia affects gene expression. They used a Myc-driven murine liver cancer model grown in 2D monolayer culture in normoxia and hypoxia as well as cells grown as 3D spheroids and performed CRISPR-based genome-wide KO screen to identify genes that play important roles in cell fitness. Some context-specific gene effects were further validated by in-vitro and in-vivo gene KO experiments.Strengths:The key findings in this manuscript are:(1) Close to 50% of differentially expressed genes were common between 2D Hypoxia and 3D spheroids conditions but they had differences in chromatin accessibility.(2) VHL-HIF1a pathway had differential cell fitness outcomes under 2D normoxia vs 2D hypoxia and 3D spheroids.(3) Individual components of the mitochondrial respiratory chain complex had contrasting effects on cell fitness under hypoxia.(4) Knockout of organogenesis or developmental pathway genes led to better cell growth specifically in the context of 3D spheroids and knockout of epigenetic modifiers had varied effects between 2D and 3D conditions.(5) Another key program that leads to cells fitness outcomes in normoxia vs hypoxia is the lipid and fatty acid metabolism.(6) Prmt5 is a key essential gene under all growth conditions, but in the context of 3D spheroids even partial loss of Prmt5 has a synthetic lethal effect with Mtap deletion and Mtap is epigenetically silenced specifically in the 3D spheroids.

We appreciate this reviewer for acknowledging the strengths of our study.

Issues to address:(1) The authors should clarify the link between the findings of the enrichment of TGFb-SMAD signaling REACTOME pathway to the findings that knocking out TGFb-SMAD pathway leads to better cell fitness outcomes for cells in the 3D growth conditions.

We have clarified this link in abstract by saying “Notably, multicellular organogenesis signaling pathways including TGFb-SMAD, which is upregulated in 3D culture, specifically constrict the uncontrolled cell proliferation in 3D while inactivation of epigenetic modifiers (*Bcor*, *Kmt2d*, *Mettl3* and *Mettl14*) has opposite outcomes in 2D vs. 3D:

(2) Supplementary Figure 4C has been cited in the text but doesn't exist in the supplementary figures section.

Sorry for this typo. It should be 5C which is Figure 2-figure supplement 3C in the new version of MS. We have corrected it now.

(3) A small figure explaining this ABC-Myc driven liver cancer model in Supplementary Figure 1 would be helpful to provide context.

We appreciate this suggestion. We have added a cartoon as Figure 1-figure supplement 1A to indicate the procedure for generation of this model.

(4) The method for spheroids formation is not found in the method section.

We described the method in our previous publication (Nature Communications 2023 Jul 6;14(1):4003.). However, we have added the information in method now, and the procedure is very simple (line 623-624). We found the murine liver cancer cell lines can readily form spheroids when they are cultured in low-attachment dish with standard DMEM complete media.

(5) In Supplementary Figure 1b, the comparisons should be stated the opposite way - 3D vs 2D normoxia and 2D-Hypoxia vs 2D-Normoxia.

We have made correction in the Figure legend of Figure S1B which is Figure 1B now in the new version of MS.

(6) There are typos in the legend for Supplementary Figure 10.

We have checked the typos.

(7) Consider putting Supplementary Figure 1b into the main Figure 1.

We have moved both Supplementary Figure 1a and 1b into main Figure 1 as Figure 1A and 1B. Hopefully, this will help the readers to catch the information easily.

(8) Please explain only one timepoint (endpoint) for 3D spheroids was performed for the CRISPR KO screen experiment, while several timepoints were done for 2D conditions? Was this for technical convenience?

As this reviewer speculated, indeed this was for technical convenience. We found that it was technically challenging to split the spheroids for CRISPR screening.

(9) In line 372, it is indicated that Bcor KO (Fig 5e) had growth advantage - this was observed in only one of the gRNA -- same with Kmt2d KO in the same figure where there was an opposite effect. Please justify the use of only one gRNA.

We actually used 4 gRNAs for each gene. In the heatmap, although one of the gRNA for each gene showed some levels of enrichment under hypoxic 2D condition, they were all highly enriched in 3D.

(10) Why was CRISPR based KO strategy not used for the PRMT5 gene but rather than the use of shRNA.? Note that one of the shRNA for PRMT5 had almost no KO (PRMT5-shRNA2 Figure 7B) but still showed phenotype (Figure 7D) - please explain.

We used shRNA as second approach for cross-validation. We agreed that the knockdown efficiency of shRNA2 was not as good as the others, with only about 40% knockdown efficiency.

(11) In Figure 7D, which samples (which shRNA group) were being compared to do the t-test?

The comparisons were for shCtrl and each of the shPRMT5. We have clarified this in figure legend.

(12) In line 240, it is stated that oxphos gene set is essential for NEJF10 cell survival in both normoxia and hypoxia conditions. But shouldn't oxphos be non-essential in hypoxia as cells move away from oxphos and become glycolytic?

This is a great question. While indeed hypoxia may promote the switch from oxphos to glycolysis, several studies showed that the low oxygen concentrations in hypoxic regions of tumors may not be limiting for oxphos, and ATP is generated by oxphos in tumors even at very low oxygen tensions (please see review Clin Cancer Res (2018) 24 (11): 2482–2490.). We therefore speculated that NEJF10 cells were still dependent on oxphos for ATP production under hypoxia. However, this needs further investigation. We have added this discussion in our manuscript (line 250-254).

(13) In line 485 it is mentioned that Pmvk and Mvd genes which are involved in cholesterol synthesis when knocked out had a positive effect on cell growth in 3D conditions and since cholesterol synthesis is essential for cell growth how does this not matter much in the context of 3D - please explain.

We thank this reviewer for this note. It seemed that only two gRNA for each were upregulated in 3D and it could be due to technical issue or clonal selection. We have deleted this sentence in our new version of MS.

**Reviewer #3 (Public review):**
Summary:In this study, Fang et al. systematically investigate the effects of culture conditions on gene expression, genome architecture, and gene dependency. To do this, they cultivate the murine HCC line NEJF10 under standard culture conditions (2D), then under similar conditions but under hypoxia (1% oxygen, 2D hypoxia) and under normoxia as spheroids (3D). NEJF10 was isolated from a marine HCC model that relies exclusively on MYC as a driver oncogene. In principle, (1) RNA-seq, (2) ATAC-seq and (3) genetic screens were then performed in this isogenic system and the results were systematically compared in the three cultivation methods. In particular, genome-wide screens with the CRISPR library Brie were performed very carefully. For example, in the 2D conditions, many different time points were harvested to control the selection process kinetically. The authors note differential dependencies for metabolic processes (not surprisingly, hypoxia signaling is affected) such as the regulation and activity of mitochondria, but also organogenesis signaling and epigenetic regulation.Strengths:The topic is interesting and relevant and the experimental set-up is carefully chosen and meaningful. The paper is well written. While the study does not reveal any major surprises, the results represent an important resource for the scientific community.

We thank this reviewer for his/her positive comments.

Weaknesses:However, this presupposes that the statistical analysis and processing are carried out very carefully, and this is where my main suggestions for revision begin. Firstly, I cannot find any information on the number of replicates in RNA- and ATAC-seq. This should be clearly stated in the results section and figure legends and cut-offs, statistical procedures, p-values, etc. should be mentioned as well. In principle, all NGS experiments (here ATAC- and RNA-seq) should be performed in replicates (at least duplicates, better triplicates) or the results should be validated by RT-PCR in independent biological triplicates. Secondly, the quantification of the analyses shown in the figures and especially in the legends is not sufficiently careful. Units are often not mentioned. Example Figure 4a: The legend says: 'gRNA reads' but how can the read count be -1? I would guess these are FC, log2FC, or Z-values. All figure legends need careful revision.

Based upon the reviewer’s suggestions, we have added details about the replicates in figure legend. For gRNA read heatmap, the scale bar indicates the Z score. We have added the information in figure legends.

Furthermore, I would find a comparison of the sgRNA abundances at the earliest harvesting time with the distribution in the library interesting, to see whether and to what extent selection has already taken place before the three culture conditions were established (minor point).

This is great point. Unfortunately, we did not perform such an analysis.

**Recommendations for the authors:**

**Reviewing Editor:**
There are three general issues:First, there is a lack of detail regarding much of the analysis. In some cases, this makes it difficult to assess the value of the data, albeit, there is generally a consensus the information is really interesting.Second, the findings - although provocative - lack mechanistic details and are focused more on descriptive findings. Hence, the manuscript would be improved by some effort at evaluating identified programs and providing some suggestions of mechanisms.Third, the authors need to put much more effort into the clarity and tightness of the presentation.

We have made clarification in response to the reviewer’s comments.

**Reviewer #1 (Recommendations for the authors):**
Figure S1C. the labeling of the lower x-axis is inverted.

Due to space limitation, we changed the figure orientation in our old version of MS. We have tilted the figure back in the new version, which is Figure 1-figure supplement 1B now.